# Recrystallization processes, microstructure and CPO evolution in polycrystalline ice during high temperature simple shear

Baptiste Journaux[1,2], Thomas Chauve[2,3], Maurine Montagnat[2], Andrea Tommasi[4], Fabrice Barou[4], David Mainprice[4], and Léa Gest[2]

[1]Department of Earth and Space Sciences, NASA Astrobiology Institute, University of Washington, Seattle, USA
[2]Université Grenoble Alpes, CNRS, IRD, G-INP, IGE, F-38000 Grenoble, France
[3]PGP, Department of Geoscience, University of Oslo, Norway
[4]Géosciences Montpellier, Université de Montpellier/CNRS, F-34095 Montpellier, France

**Correspondence:** Baptiste Journaux (baptiste.journaux@gmail.com), Maurine Montagnat (maurine.montagnat@univ-grenoble-alpes.fr)

**Abstract.**

Torsion experiments were performed in polycrystalline ice at high temperature ($0.97\ T_m$) to reproduce the simple shear kinematics that are believed to dominate in ice streams and at the base of fast flowing glaciers. As clearly documented more than 30 years ago , under simple shear ice develops a two-maxima c-axis Crystallographic Preferred Orientation (CPO), which evolves rapidly into a single cluster CPO with c-axis perpendicular to the shear plane. Dynamic recrystallization mechanisms that occur in both laboratory conditions and naturally deformed ice are likely candidates to explain the observed CPO evolution. In this study, we use Electron Backscatter Diffraction (EBSD) and Automatic Ice Texture Analyzer (AITA) to characterize the mechanisms accommodating deformation, the stress and strain heterogeneities that form under torsion of an initially isotropic polycrystalline ice sample at high temperature, and the role of dynamic recrystallization in accommodating these heterogeneities. These analyses highlight an interlocking microstructure, which results from heterogeneity-driven serrated grain boundary migration, and sub-grain boundaries composed by dislocations with $[c]$-component Burgers vector, indicating that strong local stress heterogeneity develops in particular close to grain boundaries, even at high temperature and high finite shear strain. Based on these observations, we propose that nucleation by bulging, assisted by sub-grain boundary formation and followed by grain growth, is a very likely candidate to explain the progressive disappearance of the c-axis CPO cluster at low angle to the shear plane and the stability of the one normal to it. We therefore strongly support the development of new polycrystal plasticity models limiting dislocation slip on non-basal slip systems and allowing for efficient accommodation of strain incompatibilities by an association of bulging and formation of sub-grain boundaries with a significant $[c]$-component.

## 1 Introduction

Ice deforms by shear in many natural conditions such as glaciers and ice sheets, and in particular along ice streams. In most of the deep ice cores studied, compression and extension are the dominant deformation geometries owing to the core location close to geographical domes or along ridges. Simple shear occurs in the deeper portions of these cores. However, the strongest

simple shear occurs in fast flowing glaciers and ice streams (see review by Hudleston (2015)). During large scale ice flow, deformation induces development of strong crystallographic preferred orientation (CPO) or texture. Since the late 70's a growing number of studies have provided measurements of increasing accuracy of the CPO evolution along ice cores (refer to Gow and Williamson (1976); Alley (1988); Lipenkov et al. (1989) for pioneer work, and Faria et al. (2014b) for a review). Pioneer work of Hudleston (1977) analyzed the CPO evolution in a narrow shear zone of the Barnes Ice Cap (Canada), providing well-documented observations of CPO development owing to shear in natural conditions.

CPO development in ice is related to the strong viscoplastic anisotropy of this material (Alley, 1992). Ice Ih is hexagonal with Space group $P6_3/mmc$, No.194 and Laue class 6/mmm. Plastic strain in ice Ih is mainly accommodated at high temperature by dislocation glide on the basal plane through three 1/3 $<11\bar{2}0>$(0001) equivalent slip systems (Duval et al., 1983; Hondoh, 2000). The viscoplastic anisotropy of the ice crystal results in strong strain heterogeneities at the inter and intra-granular scale when polycrystalline ice is submitted to a macroscopic stress (Duval et al., 1983; Grennerat et al., 2012, and references therein) and hence in activation of dynamic recrystallization processes, even at very low strains (Duval, 1979; Duval et al., 1983; De La Chapelle et al., 1998; Chauve et al., 2017a).

In return, the CPO-induced anisotropy at the macro-scale leads to a bulk anisotropic viscosity of ice that impacts ice rheology. This macro-scale mechanical anisotropy is responsible for modifications in ice stratigraphy and age-depth relationship (see e.g. (Martín et al., 2009; Buiron et al., 2011)). It may also produce large scale flow heterogeneities, such as basal folding (see e.g. Dahl-Jensen et al. (1997); Thorsteinsson and Waddington (2002); Gillet-Chaulet et al. (2006); Ma et al. (2010); Bons et al. (2016)). In simple shear the evolution of CPO (as well as other factors like the grain size) is associated with an increase in shear strain rate by a factor of ten at constant stress (Cuffey and Paterson, 2010; Hudleston, 2015).

This phenomenon is well-studied in the Earth mantle, where olivine CPOs produced by similar deformation and recrystallization mechanisms (cf. reviews by Ismaïl and Mainprice (1998); Tommasi and Vauchez (2015)) result in elastic and viscoplastic anisotropy. The elastic anisotropy in the mantle may be directly observed in seismic data (cf. Tommasi and Vauchez (2015)). The viscoplastic anisotropy is only observed indirectly, via the changes it produces in the deformation patterns. It plays nevertheless a major role in the development of plate tectonics (Tommasi et al., 2009).

In addition to the obvious implications of ice rheology studies to ice caps and glaciers dynamics, ice is also often used as a reference material to study anisotropic mechanical behaviors in rocks and alloys. Since ductile deformation experiments in rocks are challenging, ice is a good analogue to polycrystalline geo-materials, since "high temperature" deformation in ice at $T/T_m$ close to one can be achieved in a cold room and at ambient pressure. Mantle rocks deformed close to the melting point in the asthenosphere beneath mid-ocean ridges, sampled in ophiolites, display strong olivine CPO due to deformation and dynamic recrystallization with fast grain boundary migration (e.g. Cassard et al. (1981); Boudier and Coleman (1981); Higgie and Tommasi (2012)). The mechanical anisotropy due to these CPOs may significantly change the flow patterns beneath the ridge (Blackman et al., 2017).

Pioneering experimental studies have shown that at low shear strains ($\gamma < 0.6$), an initially random CPO polycrystalline ice sample develops a bimodal distribution of the grains c-axes (Kamb, 1972; Duval, 1981; Bouchez and Duval, 1982). These are noted as the M1 sub-maximum, which remains normal to the macroscopic shear plane, and the M2 sub-maximum, which

rotates from near parallel to the shear plane towards the M1 maximum with increasing finite shear strain (Bouchez and Duval, 1982). For important shear strains ($\gamma \geq 2$), only the M1 maximum persists. This orientation corresponds to crystals with their basal planes well oriented for dislocation glide parallel to the shear plane. A similar evolution was observed in more recent shear experiments on artificial ice polycrystals by Li et al. (2000) and Budd et al. (2013), as well as by Wilson and Peternell (2012) which analyzed the influence of the initial CPO and the importance of recrystallization processes on the CPO evolution. The vanishing of the M2 maximum has been hypothesized by Bouchez and Duval (1982) to results from rigid body rotation and recrystallization of grains unfavorably oriented for slip.

A large part of the knowledge on the microscopic processes occurring in polycrystalline ice under simple shear deformation is still derived from data published over 30 years ago (Kamb, 1959, 1972; Duval, 1981; Bouchez and Duval, 1982; Burg et al., 1986). The tools and methods used in these studies to analyze the CPO were often manual and highly dependent on the operator experience. Electron Back-Scattered Diffraction (EBSD) and Automatic Ice CPO Analyzer (AITA) can now provide high spatial and angular resolution quantitative data, enabling a global and statistical study of the processes accommodating strain at the micro-scale. Recent experiments (Qi et al., 2017, 2019) using these new characterization techniques have shed new light in some aspects of the question. They have, for instance, disproven the hypothesis by Kamb (1972) that CPO evolution in ice mainly depends on the finite shear strain and is not sensitive to temperature, strain rate, or stress. Indeed, Qi et al. (2017) showed that during axial compression the final CPO is sensitive to stress or strain rate and Qi et al. (2019) that the rate of evolution of the CPO in simple shear is sensitive to temperature.

A modern and comprehensive update of the observations of ice deformation in simple shear seems more and more necessary as the increase of calculation power allows to run mean-field or full-field polycrystal plasticity models routinely (see Montagnat et al. (2014) for a review and Llorens et al. (2017) for a recent application). These new numerical approaches require accurate experimental constraints on the microstructure evolution. It is worth noting that these models are still unable to accurately reproduce double sub-maximum of the c-axis preferred orientation evolution of polycrystalline ice observed in simple shear experiments (Bouchez and Duval, 1982; Burg et al., 1986; Wilson and Peternell, 2012) and natural shear zone (Hudleston, 1977).

The complex interplay of dislocation slip system activities, their interactions, and the role of the various mechanisms accommodating the local strain and stress heterogeneities, such as cross-slip, kink-banding, and dynamic recrystallization, is still either not or poorly represented in most of the existing modeling approaches, very likely because of the complexity of the interactions between these mechanisms. Concerning the olivine CPO development in the mantle, Signorelli and Tommasi (2015) succeeded to reproduce the evolution of olivine CPO in simple shear with a mean-field modeling approach, by making use of a two-level mechanical interaction scheme that integrates the formation of low-angle grain boundaries as expected during dynamic recrystallization.

The present work aims at providing new constraints on the processes controlling CPO evolution in simple shear in polycrystalline ice by presenting a comprehensive description of the evolution of CPO and grain shape fabric in polycrystalline ice during hot torsion experiments. We use state of the art analytical techniques: EBSD equipped with a cryostage and an Automatic Ice Texture Analyzer (AITA), to retrieve quantitative data of unprecedented spatial and angular resolution on crys-

tallographic orientations, grain size, and grain shape fabrics as a function of shear strain. These data then are used to estimate the active dislocation systems and to investigate the role of sub-grain scale processes on the recrystallization regime that allows the formation of a CPO and microstructure more favorable to shear strain.

## 2 Experimental methods

### Polycrystalline ice preparation

Unstrained equiaxial polycrystalline ice was prepared by evenly packing 200 $\mu$m sieved ice particles in a mould. The container was sealed and the air contained in the porosity was removed using a primary vacuum rotary vane oil pump. Outgassed water at 0°C was quickly introduced afterwards to fill the porosity. The resulting mush was frozen bottom up at -5°C for 48 hours. To enable a more homogenous grain size and recovery of any small strain accumulated during grain growth, the granular ice cylinders were annealed for at least 120 h at -7°C. This method produces untextured homogenous samples with negligible porosity and ∼1.5 mm mean grain size. Ultra pure Milli-Q™ water (18 MΩ·cm) was used for all sample preparations. Cylinders around 35 mm in diameter and 60 mm in height were carved from the granular ice using a lathe located in a -10°C cold room. Specific sample sizes are indicated in Table 1.

### Torsion experiments

Deformation experiments were performed using a torsion apparatus placed inside a cold room at a temperature of -7±0.5°C ($0.97 \cdot T_m$, with $T_m$ the melting temperature at 0°C ). Samples were carefully fixed in the apparatus ensuring no axial load. The design of the torsion apparatus does not allow for displacements parallel to the rotation axis; the imposed deformation is therefore perfect simple shear. During the experiments, the evolution of the CPO under these fixed-end boundary conditions might produce axial stresses (Swift, 1947; Li et al., 2000). The latter cannot be measured in the present apparatus, but polycrystal plasticity models indicate that these axial stresses may attain values similar to those of the shear stresses when the CPO is oblique to the imposed shear (Castelnau et al., 1996). A more precise description of the apparatus is provided in Supplementary material. Straight lines parallel to the rotation axis were traced on the surface of the sample to allow for observation of any strain heterogeneity along the cylinder (Fig. 1). The surface of the sample was coated with silicon grease and wrapped gently with a cellophane foil to ensure negligible loss by sublimation.

We imposed a constant torque $M$ on one site of the sample corresponding to a maximum stress $\tau_{max}$ between 0.4 and 0.6 MPa (Table 1). The $\tau_{max}$ was calculated from the torque $M$ using a stress-strain power law as described in Paterson and Olgaard (2000). Since most of the present experiments recorded secondary creep conditions (Fig. 1), stress exponent of $n = 3$ was chosen based on results from Duval et al. (1983) and more recent work from Treverrow et al. (2012) that observed stress exponents between 2.9 and 3.1 during the secondary creep regime, and of 3.5 for tertiary creep in compression and shear tests. The actual torque applied to each sample are indicated in Table 1. These values of maximum tangential stress have been chosen because it allows high enough strain rates, while preventing the formation of cracks.

Macroscopic strain (Fig. 1)) was followed using a Linear Variable Differential Transformer (LVDT) during the primary creep regime (i.e. small strains) and a home-made optical rotary encoder device (5000 ipr) for larger strains. Maximum macroscopic shear strain $\gamma$, the maximum natural extensional strain $\varepsilon$, and the strain rate $\dot{\varepsilon}$, calculated through $\varepsilon = \ln \sqrt{\frac{1}{2} \left( 2 + \gamma^2 + \gamma \sqrt{4 + \gamma^2} \right)}$ (Paterson and Olgaard, 2000), are reported in Table 1.

**Analytical methods**

CPO, microstructure, and sub-grain strain heterogeneity analyses were performed using both AITA and EBSD, as the combination of the two techniques allows the acquisition of a comprehensive dataset. AITA enables the analyses of large samples (up to $\sim$10x10 cm$^2$), but only recovers the c-axis (optical axis) crystallographic orientations. On the other hand, the cryo-EBSD setup used in the present study (CamScan X500FE CrystalProbe at Geosciences Montpellier) is only capable of mapping smaller
ice samples of $\sim$1x2 cm$^2$, but is able to recover the full orientation of the crystal (all crystallographic axes) with an expected angular resolution better than 0.7° (Randle, 1992) and with a practical spatial resolution for ice down to 5 $\mu$m . Sections tangential to the experimental cylinders were cut out for both AITA and EBSD analyses. The general geometry of these sections is illustrated in Fig. 2. To provide flat surfaces for AITA thin sections and EBSD samples, the sample sections were microtomed by less than 1 mm, which gives an error on $\gamma_{max}$ of less than 5 %.
AITA has been performed on thin sections typically 300 $\mu$m thick, as it is routinely done on ice (Russell-Head and Wilson, 1999; Montagnat et al., 2011; PETERNELL et al., 2011; Chauve et al., 2015; Wilson et al., 2015). We worked at a spatial resolution of 25 $\mu$m and were able to obtain an angular resolution of about $\sim 3$°. All optical CPO measurements were conducted at -7°C. Grains boundaries (GB) were extracted using the segmentation technique described in Montagnat et al. (2015) which automatically defines grain boundaries using color change detection in AITA maps, followed by manual corrections based on
coherent microstructure shapes and discernible misorientations. All manual corrections are operator dependent, so all analyses have been performed by the first author for consistency. This allowed us to differentiate, with a good level of confidence, sub-grain boundaries (SGB) from grain boundaries, even if optical measurements do not give access to the full crystallographic orientation, as EBSD measurements do. From the extracted skeleton images we derived grain size and shape statistics using the PolyLX Matlab® toolbox developed by Lexa (2003). Grain size frequency plots are reported for every finite shear strain
in Supplementary material. From the grain shapes, we calculated the PARIS grain shape factor to quantify grain boundary interlocking as PARIS = 2·(P-PE)/PE*100%, with P the perimeter and PE the convex hull perimeter (Heilbronner and Barrett, 2014). This parameter increases with increasing grain boundary sinuosity.

Electron backscattered diffraction (EBSD) patterns were obtained with a CamScan X500FE CrystalProbe equipped with a liquid Nitrogen cryogenic stage (GATAN®) at Geosciences Montpellier (France). Sample surfaces were prepared by carefully
flattening the surface with a microtome blade at -60°C inside a freezer placed near the EBSD just before loading it quickly on the cold stage. This method provides a very good surface finish with no significant frost during measurements and limits to a minimum the heat transfer to the sample, ensuring limited microstructure evolution due to annealing. The CrystalProbe-EBSD working conditions were 15 kV, 3.5 nA, and low vacuum with 1 Pa of gaseous Nitrogen and 25 mm working distance. More details on the EBSD geometry can be found in Demouchy et al. (2011). To avoid ice samples sublimation in the vacuum

chamber, which happens at approximately -60°C at 1 Pa for ice Ih (Fernicola et al., 2011), the temperature of the cold stage was kept at $-100 \pm 10$°C during the analyses. EBSD Kikuchi patterns were automatically indexed using the HKL Channel5™ software suite (Oxford Instruments®). Maps were acquired with a spatial resolution of 25 $\mu$m identified as a good compromise between resolution and time constrains, with an indexation rate higher than 85%. Analyses of the EBSD maps were made using the MTEX Matlab® toolbox (Hielscher and Schaeben, 2008; Bachmann et al., 2010, 2011; Mainprice et al., 2014).

To observe the geometry of sub-grain boundaries and extract statistical quantities, local misorientation analyses, such as Kernel average misorientation (KAM) or the local misorientation relative to the mean orientation (Mis2Mean), were performed on the EBSD data (see review by Wright et al., 2011). Both analyses enable visualization of the spatial distribution of intragranular stored strain structures, generally in the form of sub-grain boundaries (Wright et al., 2011), but do not provide any information about the type of dislocations accommodating the observed misorientation. On the other hand, the Weighted Burgers Vector (WBV) analysis allows to extract a minimum value for the dislocation density in sub-grain boundaries and provides an estimate of the slip systems necessary to explain the observed misorientations (Wheeler et al., 2009). As conventional EBSD maps are 2D, only 5 components of the Nye tensor $\alpha$ can be extracted ($\alpha_{12}, \alpha_{21}, \alpha_{13}, \alpha_{23}, \alpha_{33}$). This allows to obtain a projection of the Nye tensor on the EBSD map plane which constrain partially the Burgers vector. It provides a lower bound for the dislocation density and constrains the Burgers vector of geometrically necessary dislocations (GND) accommodating the observed misorientations. Even if WBV analysis does not recover the full Burger vectors coordinates, it does not provide "false" components. This means that if a $[c]$-component is deduced from WBV analysis of a 2D EBSD map, the true Burgers vector does actually contains at least this amount of $[c]$-component. For the WBV analysis we used the Matlab® toolbox developed by Wheeler et al. (2009) and previously used on ice in Chauve et al. (2017b). To provide the relative amount of $[c]$-component dislocations, the calculated WBV for each pixel is projected on the non-independent 4 hexagonal lattice directions ($[11\bar{2}0]$, $[\bar{2}110]$, $[1\bar{2}10]$, $[0001]$), respectively called $WBV_{a1}$, $WBV_{a2}$, $WBV_{a3}$ and $WBV_c$. The ratio of $[c]$-component ($WBV_c$) is then given by the relation $rWBV_c = (|WBV_c|/||\text{WBV}||)$, with $||\text{WBV}||$ the Euclidian norm of the WBV. A high-pass value of 0.4° ($||\text{WBV}|| > 2.8 \cdot 10^{-4} \mu m^{-1}$) was imposed for the misorientation analysis to filter the noise resulting from the angular resolution of the EBSD data. We suggest to the interested readers to refer the to Appendix A of Chauve et al. (2017b) for more details about this analysis. Nevertheless, it should be recalled that the WBV analysis does not directly provide information about mobile dislocations, responsible for the majority of the plastic deformation. It only measures the GND, as it is based upon the Nye tensor.

## 3 Results

Five torsion experiments were performed up to different finite shear strains $\gamma_{max}$, from 0.012 to 1.94. Data for all experiments are presented in Table 1. One sample (TGI0.71), with a $\gamma_{max}$ of 0.71, was annealed for 72 hours at -7° after the deformation experiment to study the effect of annealing on both CPO and microstructure. The microstructure of one of these samples has been partially described in Chauve et al. (2017b) to compare the statistical representation of WBV (or Burgers vector) with $[c]$-component in GNDs in samples submitted to uniaxial unconfined compression and torsion.

**Macroscopic strain**

Retrieved samples presented homogeneous strain with no apparent localization or cracks, as illustrated in Fig. 1. TGI0.012 and TGI0.2 shear strain data were very noisy due to signal transmission issues on the acquisition setup, which were fixed for the TGI0.42 and TGI1.96 runs. Typical primary creep regime (i.e. hardening) is observed for TGI0.42 and TGI1.96 in Fig. 1 until approximately 20,000 $s$ ($\approx$ 6 h). The noise in TGI0.012 and TGI0.2 data was too strong to distinguish any primary creep hardening. After this phase of hardening, a minimum in strain rate is reached (secondary creep regime). This minimum is clearly visible in experiments TGI0.2, TGI0.42, TGI0.71 and TGI1.96 at 200,000 $s$ ($\approx$ 56 h). For the longer TGI1.96 experiment, after the secondary creep phase, strain rate continuously increased until achieving an almost steady state strain rate of $8.5 \cdot 10^{-7}$ $s^{-1}$, suggesting that tertiary creep regime was reached for this run (Fig. 1). TGI0.012 stayed mostly in the primary creep regime with a $\gamma_{max} = 0.012$. Samples TGI0.2 and TGI0.42 experienced a maximum shear strain of 0.2 and 0.42 respectively. The final strain rate for these two experiments is illustrated by the tangent dashed line in Fig. 1, which shows that softening was still occurring when the samples were unloaded (only the last data points lies on the tangent). We therefore consider that those experiments where still in a transient regime when stopped. A strain step is visible for TGI0.42 between 550,000 s ($\approx$ 153 h) and 650,000 s ($\approx$ 180 h) ; it corresponds to a period where the weight used to apply constant torque was in contact with the metallic frame of the torsion apparatus, consequently decreasing the applied stress. After 650,000 s ($\approx$ 180 h), the weight was put back in a good configuration and the strain curve recovered its pre-step slope. The significant variations observed in strain rate evolution with time between the different runs cannot be attributed to a variation in initial grain size, CPOs or in the applied torque alone (Table 1), but rather to coarse-grained microstructure of the samples, which resulted in fewer than 25 grains per radii and, hence, in a not perfectly isotropic macroscopic behavior. The strain/time curves presented in figure 1 are therefore useful to characterize each run creep regime independently, bur should be used with care in comparison between different samples or with other experiments.

**Microstructure evolution**

AITA maps of tangential sections for each sample and associated analyses are presented in Fig. 2 and Table 2. The evolution of CPO and microstructure (grain size and shape) as a function of finite shear strain is clearly visible. The color wheel gives the orientation of the <0001> c-axis. The annealed sample TGI0.71 is discussed separately at the end of this section. As the shear strain increases, the distribution of colors changes from random color toward a predominance of red and blue grains, until the map becomes dominated by red color grains for TGI1.96 ($\gamma_{max}$=1.96).

This evolution records the progressive change in orientation of the c-axis with increasing shear strain, displayed as pole figures in Fig. 2b. For the undeformed ice sample ($\gamma_{max}$=0) we observe approximately random orientations of c-axes with a texture (CPO) index $J_i$ = 1.07. The c-axes texture (CPO) J-index is derived by calculating the second moment of the Euler angles orientation distribution function (ODF) of discrete crystal orientation data (Bunge, 1982). It is commonly used to quantify CPO strength in geology and in material science. The strain reached by TGI0.012 sample is not high enough to produce any significant change of the CPO. As the strain increases we observe the formation of two sub-maxima, as reported

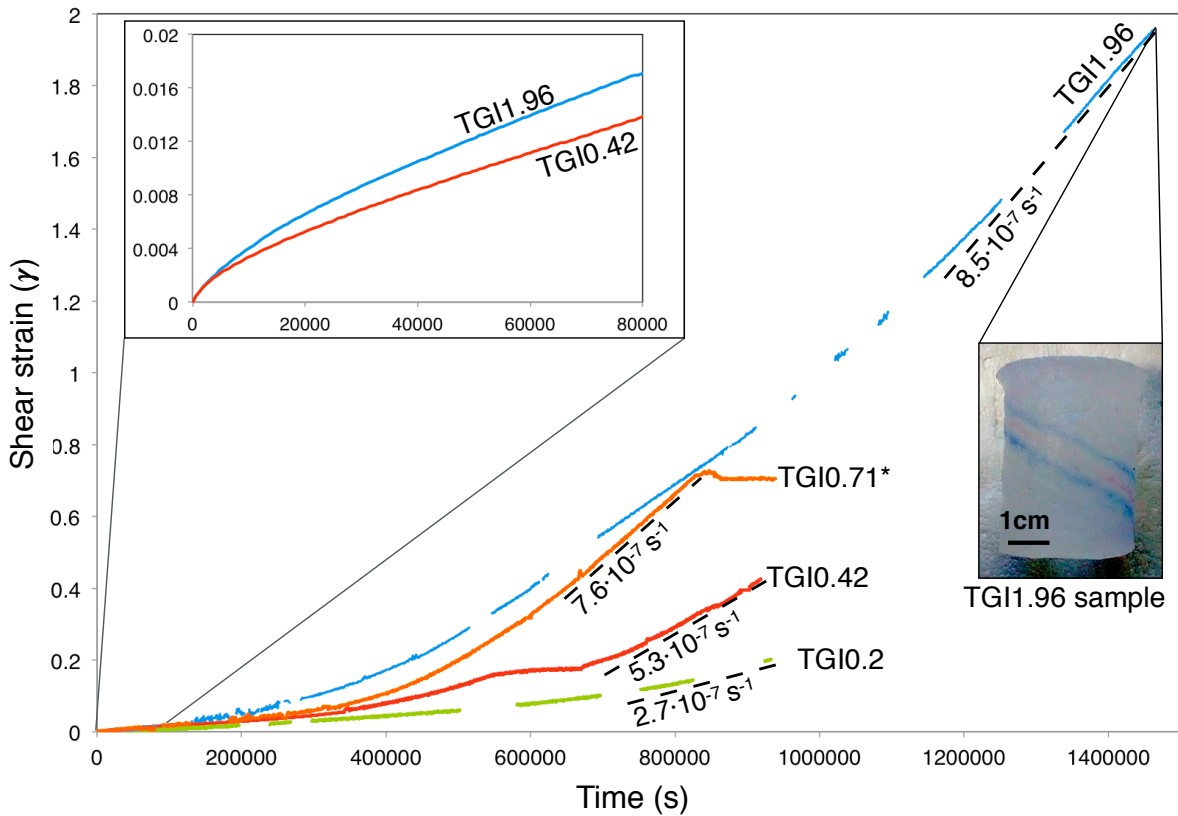

**Figure 1.** Creep curves for TGI0.2, TGI0.42, TGI0.71 and TGI1.96 experiments with the tangent to end of curve as black dashed lines, and corresponding final strain rate $\dot{\varepsilon}_f$. The blank parts of some of the curves correspond to a loss of signal between the acquisition setup and the sensor located in the cold room. In the top left frame is a zoom on the primary creep regime ($< 10$ h) for better readability. The photograph in the bottom right shows the TGI1.96 sample after unloading, with the inclined ink color stripes, initially straight, used as strain markers.

in previous studies (Kamb, 1959, 1972; Duval, 1981; Bouchez and Duval, 1982; Burg et al., 1986), one almost perpendicular to the shear plane (M1) and the other one at low angle (initially 11-17°) from the shear plane (M2) as indicated in Fig. 2. With increasing shear, M2 rotates into higher angles to the shear plane (up to 47°) while M1 remains stable at 90° from the macroscopic shear plane. More importantly, the maxima M1 and M2 increase and decrease in intensity, respectively. This
5   evolution towards a single maximum CPO results in an increase of the J-index (Fig.3). The plane containing the finite extension direction (ED) and normal to the compression direction (CD) has a very low $[c]$-axis CPO density for all $\gamma \geq 0.2$. One should note that for $0.2 \leq \gamma \leq 0.42$, the bimodal pattern formed by the two M1 and M2 submaxima is almost symmetric relatively to the maximum finite extension axis (ED).

Nevertheless the symmetry between M1 and M2 around the finite extension direction is not perfect. The angle between M2
10   and ED is generally larger than the angle between M1 and ED by 1 to 3°. The exception is the annealed sample TGI0.71, where

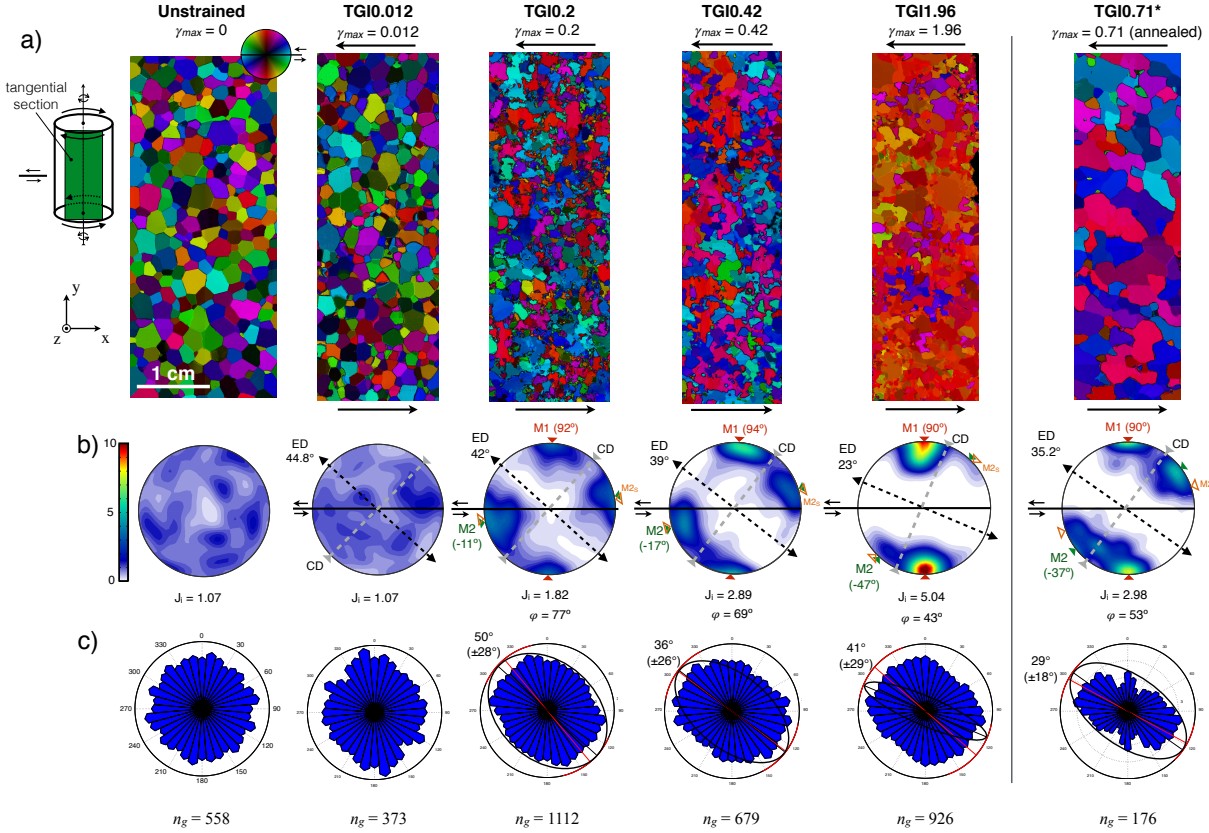

**Figure 2.** a) AITA analyses of tangential sections for every sample showing the evolution of CPO and microstructure, with $x$ parallel to the shear plane and $y$ parallel to the torsion axis. The scheme on the upper right shows the geometrical relation between the tangential section and the retrieved cylindrical samples. Arrows indicate the shear directions. Color-coded c-axis orientation is indicated by the color wheel. b) Pole figures (equal area) showing the c-axis orientations evolution with shear strain. Shear direction, direction of finite extension (ED) and compression direction (CD) are indicated with a black line, doted black line and grey dashed line, respectively. Direction angles for the M1 and M2 submaxima are reported on the pole figures with red and green triangles respectively along with their angles with the ED direction. Theoretical perfectly symmetric M2 poles to M1 with ED direction (M2$_S$) are represented in orange outline open triangles. J-index (Bunge, 1982) and $\varphi$ angle between M1 and M2 submaxima are indicated below for $\gamma_{max} \geq 0.2$. c) Rose diagram of grain shape fabrics. Mean orientation is indicated in red line and 95% confidence interval as a red arc. Finite shear strain ellipses are plotted onto each rose diagram. The total number of grains segmented is reported as $n_g$.

the difference between the two is 17.4°, with M2 closer to ED than it should have been for a perfect symmetry. This change may be due to a post-deformation CPO evolution, due to grain growth during annealing. A small lag in the reorientation of the M2 submaximum relative to the M1 submaximum is also observed in other simple shear experiments (Bouchez and Duval, 1982; Qi et al., 2019). The limited number of experiments performed in the present study precludes a statistical analysis of this

behavior. The evolution of the angle between M1 and M2 ($\varphi$) with increasing shear strain is discussed in more detail in Qi et al. (2019), who compared observations from ice shear experiments carried out at different temperatures and numerical modeling.

AITA extracted grain statistics are presented in Table 2. With increasing $\gamma$ the median grain size (with associated interquartile range) diminishes from 0.7(4) mm to 0.12(16)mm. The median is preferred relative to the mean because of the non-normal distribution of grain sizes. The coarser grain size in TGI0.42 sample compared to TGI0.2 and TGI0.012 is probably due to a

difference in the initial grain size, but the grain sizes remain comparable if interquartile ranges are considered. Grain shape fabrics calculated using moment-based ellipse fitting technics (Lexa, 2003) are illustrated as rose diagrams in Fig. 2c. As the shear strain increases, we observe the development of a grain shape fabric following the maximum finite elongation direction indicated with the corresponding overlying strain ellipse in Fig. 2.c. The grain shape fabric clearly evolves until $\gamma = 0.42$ and then does not vary significantly up to $\gamma = 1.96$

Grain boundary interlocking estimated through the PARIS factor (reported in Table 2) increases during the transient strain regime (from 2.2 % at $\gamma = 0$ to 11.5 % at $\gamma = 0.42$) and then decreases. At the highest shear strain of $\gamma = 1.96$, the PARIS (5.9%) is half of the one at a shear strain of 0.42.

The annealed sample TGI0.71 shows a median grain size of 1.7 mm, above the initial unstrained value of 0.7 (Table 2). The CPO does not seem to be modified by the annealing with the M1 and M2 maxima maintained and a J-index = 2.96, which lies in

between the values for $\gamma_{max}$ 0.42 and 1.96, as would be expected from a non-annealed sample with the same finite shear strain (Fig. 2b). The grain shape fabric in the annealed sample is nevertheless more marked than in the strained, but non-annealed samples. The PARIS factor decreases significantly, as much of the interlocking of grains is smoothed by annealing.

In summary, the crystallographic orientation evolution during simple shear strain of polycrystalline ice can be described as follows. M1 orientations rapidly develop (by $\gamma = 0.2$). They remain stable in orientation and increase in intensity until

they become predominant at high strains ($\gamma > 1.96$). The M2 maximum also rapidly develops; it is roughly symmetric to M1 respectively to the maximum finite shorterning direction. However the M2 maximum continuously decreases in intensity during the transient regime. The nearly complete disappearance of M2 correlates with achievement of steady state (tertiary creep). All other crystallographic orientations start to disappear as early as $\gamma = 0.2$ and this tendency is accentuated with increasing shear strain. Finally, if annealing occurs after the strain stops, the CPO is maintained even if the grain shape fabric and grain size are

modified.

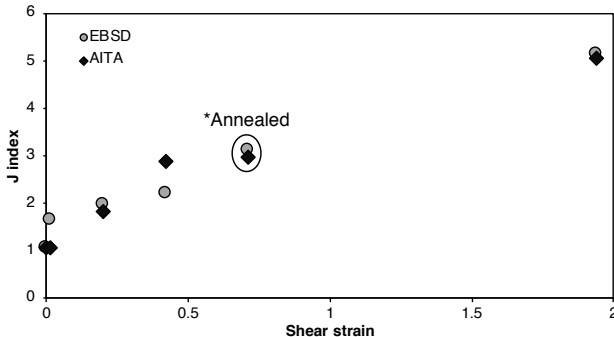

**Figure 3.** c-axes texture (CPO) index $J_i$ determined from AITA and EBSD data for all the shear strains investigated. Annealed sample TGI0.71 is highlighted for clarity.

### EBSD

### Maps & textures

Multiple EBSD maps were acquired for every final shear strain. Representatives examples of EBSD maps for each sample are shown in Fig. 4.a, with the number of maps $n_m$ acquired for each $\gamma_{max}$ indicated at the bottom. The access to the full crystallographic orientations enables more complete grain segmentation based on misorientation data. Grain boundaries are characterized by misorientations $\geq 5°$ and subgrain boundaries between $1°$ and $5°$ Chauve et al. (2017b).

The orientations of $<0001>$ $<10\bar{1}0>$ and $<11\bar{2}0>$ axes relatively to the shear plane and direction are shown in Fig. 4.b. All maps acquired for each $\gamma_{max}$ were combined for better statistical analysis. An excellent level of agreement with CPOs from AITA is observed for $<0001>$ CPOs, which develop rapidly M1 and M2 sub-maxima, with disappearance of the M2 maximum and reinforcement of the M1 one at high shear strain. The increase in c-axis CPO J-index confirms this evolution, with very consistent values with those based on the AITA measurements for a same $\gamma_{max}$ (Fig. 3). For the high shear strain sample at $\gamma_{max}$=1.96, the $<10\bar{1}0>$ and $<11\bar{2}0>$ axes ($<a>$ and $<m>$ axis) form a girdle, which tends to align in the shear plane. Within this girdle, there is a preferred orientation of both $<a>$ and $<m>$ directions toward the shear direction. The present CPO is similar to those formed in direct shear experiments (Qi et al., 2019). It is consistent with equivalent contribution of the three $<a>$ axes in accommodating shear on the (0001) plane.

Some elongation of the distribution of the M1 and M2 submaxima towards the Z direction (Figure 4.b), which is the normal to the shear direction in the shear plane, is visible in our results. This elongation is best expressed for the M1 maximum in the highest strain sample TG1.96, for which pole figures for $<0001>$ $<10\bar{1}0>$ and $<11\bar{2}0>$ lattice vectors are now represented in two perpendicular reference frame in figure 4.b for better readability. Similar elongated distributions of $<c>$ axes have been reported in direct shear experiments by Qi et al. (2019). Some elongation of the M1 maximum is also observed in the highest shear strain sample (gamma = 2) of Bouchez and Duval (1982) as well as in other shear experiments in Li et al. (2000), Wilson

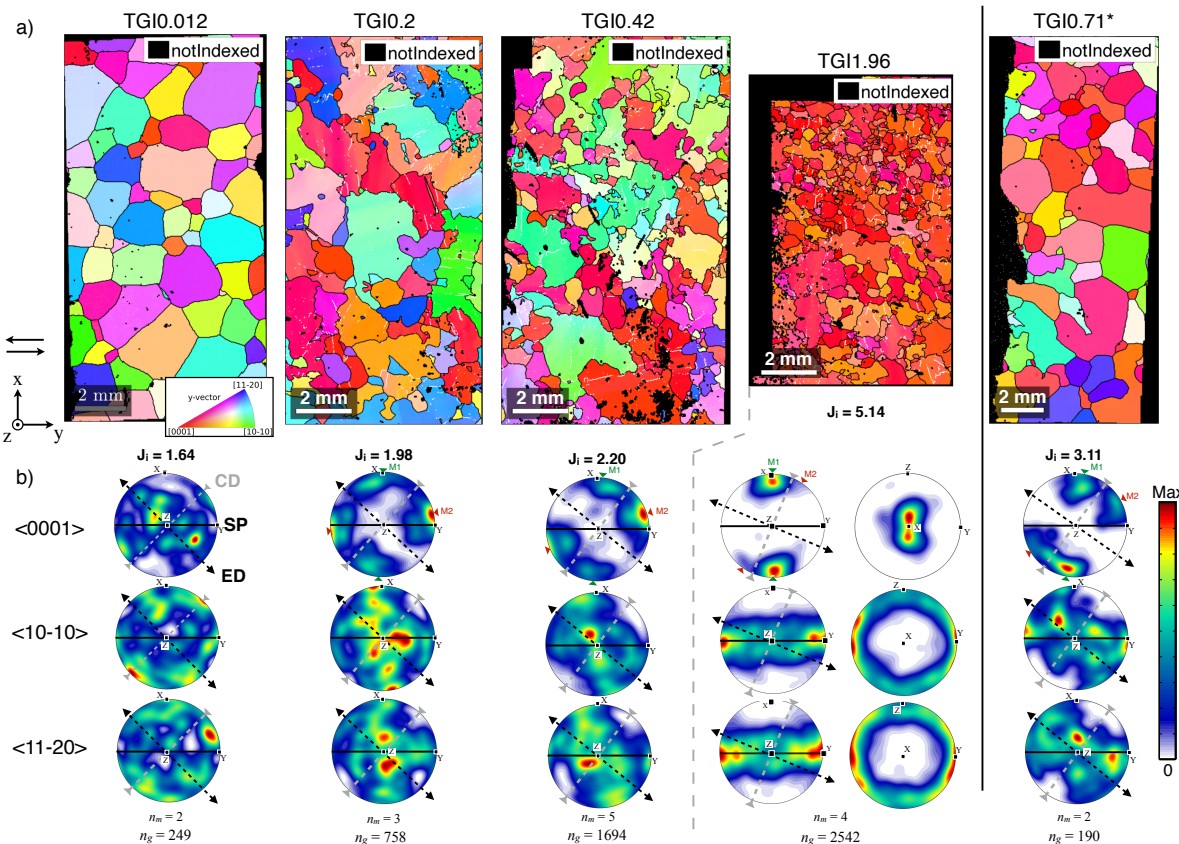

**Figure 4.** a) Example of EBSD patterns obtained from tangential sections for each maximal shear strain, at -7°C. Map colors correspond to the orientation of the crystallographic lattice toward to the y-axis (torsion axis), as indicated in the inverse pole figure on TGI0.012 EBSD map (top left). Red grains have <0001> (c-axis) parallel to the torsion axis, green grains have <10$\bar{1}$0> (a-axis) parallel to the torsion axis, and blue grains <11$\bar{2}$0> (m-axis). Boundaries with misorientation greater than 5° are represented in black lines and boundaries between 1° and 5° are depicted as white lines. b) Lower hemisphere pole figures (equal area) representing the orientations of <0001>, <10$\bar{1}$0> and <11$\bar{2}$0> directions of the EBSD data were combined for each $\gamma_{max}$. Additional pole figures projected normal to the X direction are represented for $\gamma_{max}$=1.96. Orientation density intensities are scaled to their maximum value for better readability. Associated c-axis texture (CPO) J-index is indicated above the pole figure. The shear plane (SP) is indicated as a black line, and the finite extension (ED) and compression (CD) directions, by a black and grey dashed line, respectively. The number of maps used to obtain the pole figures is indicated as $n_m$ and the total number of grains segmented as $n_g$.

and Peternell (2012) and Budd et al. (2013). However, most naturally sheared ice samples do not have elongated <c> maxima (Hudleston, 1977).

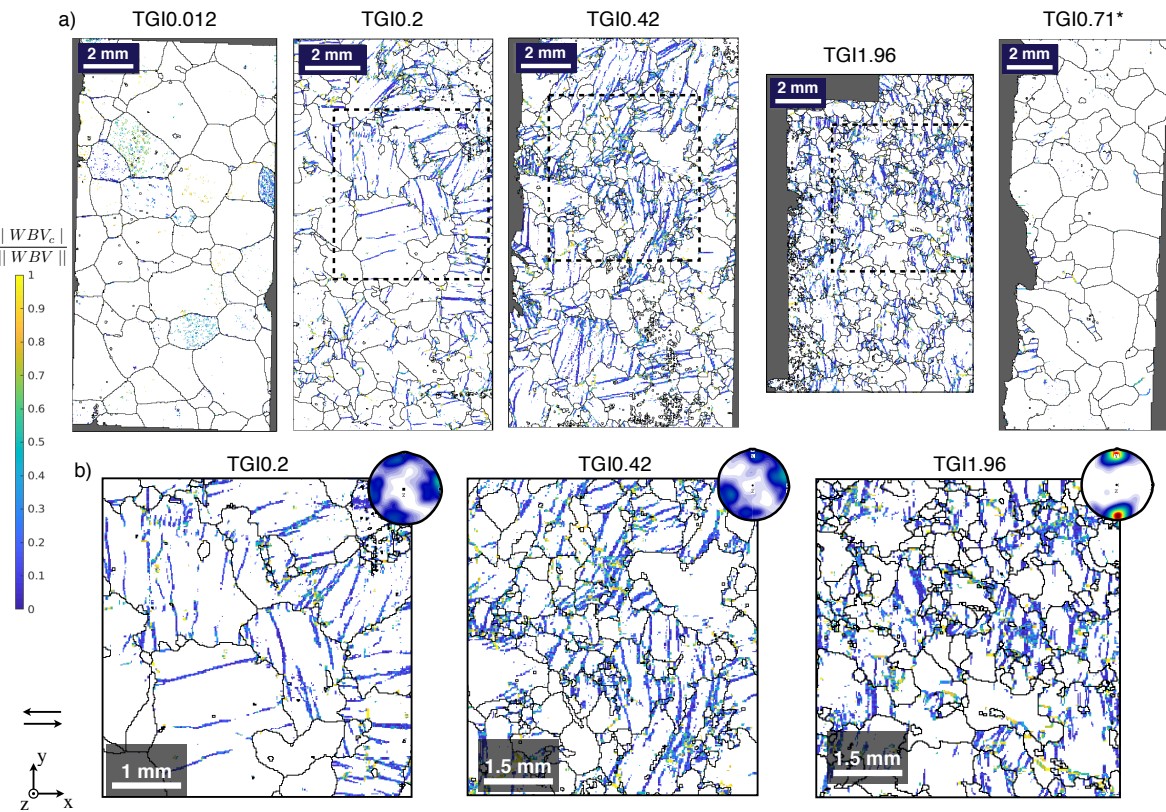

**Figure 5.** Maps of the norm of the Weighted Burgers Vector, ||WBV||. a) Map examples for every shear strain. zoomed in region in b) delimited by black dashed rectangles. b) Details on the TGI0.2, TGI0.42 and TGI1.96. Grains boundaries ($> 5°$) are reported as black lines. Non-indexed regions reported in grey. Pole figures representing c-axis orientations are reported at the top right.

## Weighted Burgers Vector analysis and intra-granular strain

During plastic strain, some of the stress on the ice lattice is relieved by the formation and glide of dislocations. Some of those dislocations are trapped in the crystal lattice, forming stable arrays of geometrically necessary dislocations (GND), which may define sub-grain boundaries. Such intragranular strain heterogeneities are observed in numerous experimental studies on polycrystalline ice and in natural environments, (Wilson et al., 1986; Mansuy et al., 2000; Piazolo et al., 2008; Montagnat et al., 2011; Weikusat et al., 2011; Faria et al., 2014a; Chauve et al., 2017b). As described in the methods section, WBV analysis allows calculation of a minimum density of GNDs and definition of the relative proportion of dislocations with $<a>$ and $[c]$-component Burgers vector within this GND population. Figure 5 provides representative ||WBV|| maps at every finite shear strain investigated.

In these WBV maps (Fig. 5) we observe a large number of linear sub-grain boundaries already developed at $\gamma_{max}$ = 0.2. Evolution of the GNDs density with increasing strain is hard to assess by comparing visually the maps for different finite shear strains, but it can be assessed by the analysis of the quantitative statistical distribution of $<a>$ and $[c]$ GNDs (Fig. 6). The sub-grain boundaries are mostly organized in walls and the direction of the trace of these planes on the thin section typically corresponds to the parent grain $[c]$-axis direction. This kind of sub-grain boundaries, very common in both natural and experimentally deformed ice, has been identified as basal tilt boundaries composed of basal edge dislocations ($b = a$, with $b$ the Burgers vector) through EBSD and X-Ray Laue diffraction (Piazolo et al., 2008; Weikusat et al., 2011; Chauve et al., 2017b). These sub-grain boundaries are visible in Fig. 5b as purple and blue lines. Other types of low angle boundaries were also identified in both natural and experimentally deformed ice; these are basal twist boundaries with basal screw dislocation and non-basal tilt boundaries with variable amounts of dislocations with $[c]$ axis Burgers vector component (i.e. $<a+c>$ and $[c]$) (Piazolo et al., 2008; Weikusat et al., 2011; Piazolo et al., 2015; Chauve et al., 2017b). Such sub-boundaries are also identified in the present study (green to yellow lines in Fig. 5b) It is interesting to note that even in the most deformed sample, which has a strong CPO with one single sub-maxima at $\gamma_{max}$ = 1.96 (at steady state tertiary creep), GNDs are very common. Both the undeformed sample (TGI0.012) and the annealed sample (TGI0.71) have much lower low angle boundary densities than the other samples. The undeformed sample map shows low angle boundaries located very close to high angle boundaries and as unconnected pixels and small segments within grains. These are best explained as low angle misindexing errors. Most grains in the annealed sample contain very few angle boundaries. In this annealed sample, only $\sim$ 4 grains contain low angle boundaries similar to the one extensively observed in the deformed but not annealed samples(Fig. 5a).

As demonstrated by Chauve et al. (2017b), the total density of GNDs (with $||WBV||$ above threshold) increases with strain (Fig. 6f). However, the ratio of $[c]$-component ($rWBV_c = |WBV_c|/||WBV||$) of GNDs remains constant at ca. 35 % ($rWBV_c > 1/3$, or $<a+c> + [c]$). For the lowest shear strain (TGI0.012) the $[c]$-component ratio is larger, but this could be due to the smaller sample size, as the density of GNDs (i.e. nb of pixels with a $||WBV|| > 2.8 \cdot 10^{-4}$ $\mu m^{-1}$) is much lower. By classifying grains by orientation, it is possible to look at the individual statistics for the grains composing the M1 and the M2 maxima and in other orientations (Fig. 6a to 6e). Analysis of the pixel orientation statistic shows that the proportion pixels with M2 orientation rapidly increases compared to the one with M1 orientation at $\gamma_{max}$ = 0.2 and then gradually decreases at $\gamma_{max}$ = 0.42, eventually disappearing at $\gamma_{max}$ = 1.96. When performing the WBV analysis for each grain orientation family, the presence of GNDs with a $[c]$-component Burgers vector does not seem to be correlated with the orientation of the grain. The only exception is the annealed case at $\gamma_{max}$ = 0.71, where recovery processes and noise due to the small sample size limit the use of such analysis. The present observations are consistent with previously published data on $[c]$-component GNDs for both shear and compression experiments from Chauve et al. (2017b).

We computed the probability density function of $rWBV_c$ as function of the distance to the grain boundary, and represented it for $rWBV_c < 1/3$, $1/3 < rWBV_c < 2/3$ and $rWBV_c > 2/3$ in Fig. 7. We are able to observe that the $rWBV_c < 1/3$ density clearly decreases around 100 $\mu$m close to the grain boundary, when both $2/3 < rWBV_c < 2/3$ and $rWBV_c > 2/3$ densities increases ($rWBV_c > 2/3$ almost doubles). This statistical analysis clearly shows the increase of $[c]$-component dislocations

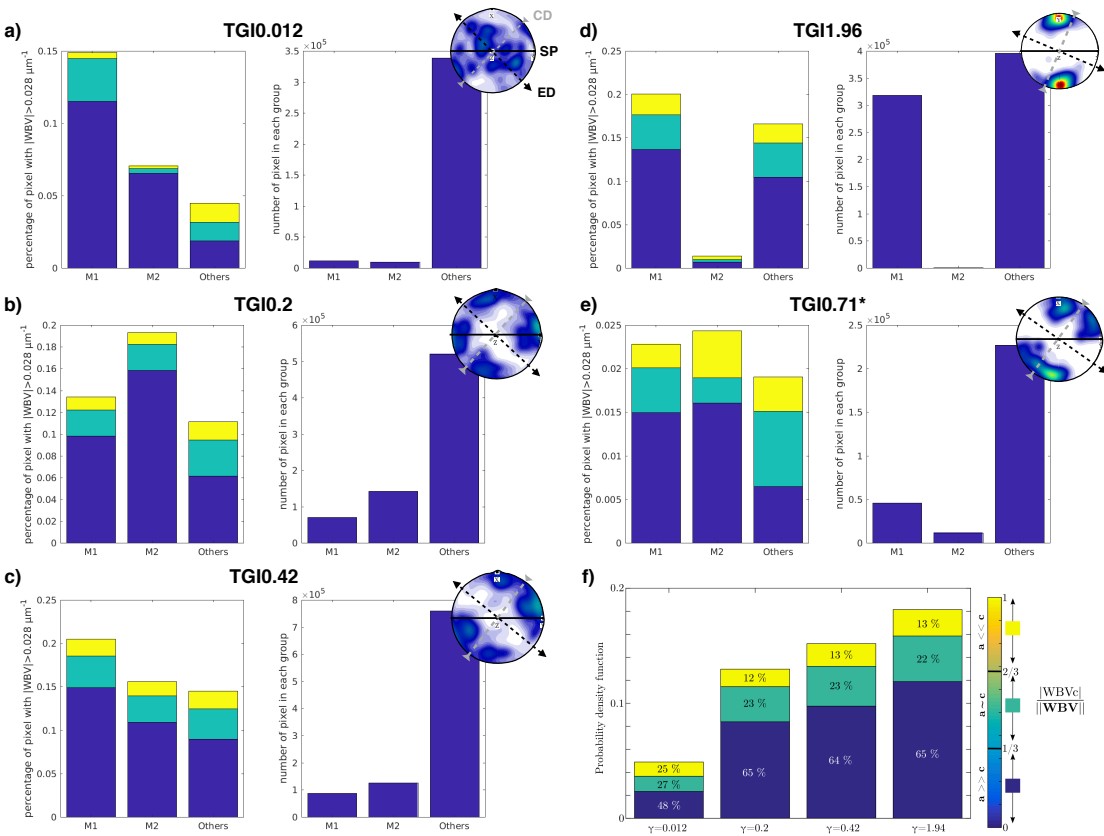

**Figure 6.** a,b,c,d,e) Statistical distribution of pixels per group of orientation (M1, M2 sub-maxima or other) and per $rWBV_c$ ensemble for each $\gamma_{max}$ 0.012, 0.2, 0.42, 1.96 and 0.71 respectively, with associated number of pixels per group. The c-axis pole figure is also reported for each finite shear strain at the top right. f) Distribution of pixels with a $||WBV||$ higher than a threshold of ($2.8 \cdot 10^{-4} \ \mu m^{-1}$). Evolution with torsion strain of the relative [c] and <a> components over the norm of the full WBV ($|WBV_c|/||WBV||$) for four distinct torsion creep tests. See Chauve et al. (2017b) for more details.

proportion in the substructures at a distance lower than 100 $\mu$m from the grain boundary. It is important to note that we do not observe any significant differences between different finite shear strains.

## 4 Discussion

As indicated by the evolution of the macroscopic strain rates on Fig. 1, we were able to analyze the CPO and microstructure
5  at different creep regimes. The near constant strain rate ($\dot{\varepsilon}_f = 8.5 \cdot 10^{-7} \mathrm{s}^{-1}$) achieved for the TGI1.96 run can be interpreted as the evidence that this sample was close to steady state in terms of mechanical behavior, typical of tertiary creep regime. The appearance and evolution of c-axis orientation sub-maxima M1 and M2 in the CPO are consistent with previous data from Hudleston (1977) and Bouchez and Duval (1982) and from the recent direct shear experiments from Qi et al. (2019). The M1

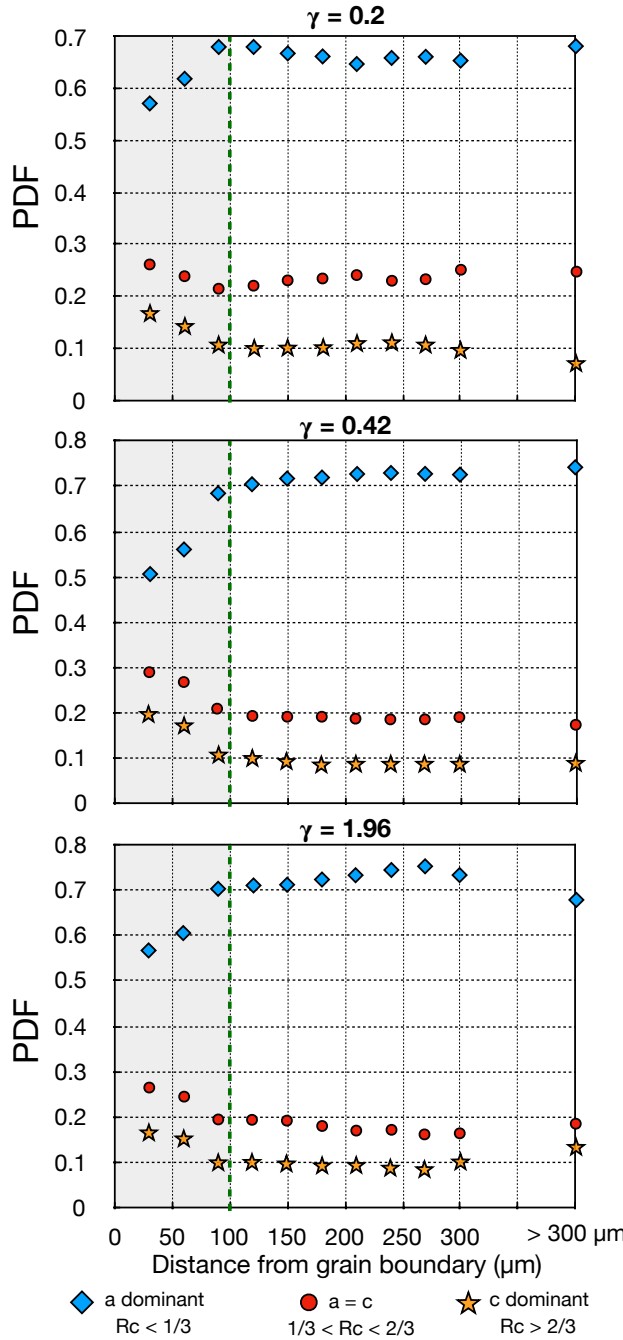

**Figure 7.** Probability density function for $rWBV_c < 1/3$ (blue diamonds), $2/3 < rWBV_c < 2/3$ (red circles) and $rWBV_c > 2/3$ (yellow stars) components of the GNDs across profiles from grain boundaries to cores of the GNDs, extracted from WBV maps, for TGI0.2, TGI0.42 and TGI1.96. The green dotted line represents the 100 $\mu$m distance to the grain boundary.

maximum remains always normal to the shear plane and the M2 sub-maximum, initially as strong as M1, progressively rotates with shear during transient creep, and decreases in intensity until it disappears, possibly as tertiary creep regime is reached.

The slow evolution of the grain shape preferred orientations and their saturation at shear strains higher than 0.2, visible from the grain shape orientation rose diagram (Fig. 2.c), could be the effect of recrystallization processes and fast grain boundary migration that counter the formation of a strong grain shape fabric. These processes may also explain why we observe a decrease in the grain boundary interlocking at high strains and even more in the annealed sample through the PARIS factor (Table 2). Both dynamic and static recrystallization should enable the appearance and growth of new grains with small PARIS factors at the expenses of older grains with serrated grain boundaries (higher PARIS). This observation underlines that the estimation of strain magnitude from grain elongation or shape is not reliable at high temperature, as noted by Burg et al. (1986). The efficiency of grain boundary migration and recovery of shape fabric is expected to significantly decrease with decreasing temperatures.

The case of the annealed sample TGI0.71 is interesting to understand how metadynamic annealing processes like grain boundary migration driven by dislocation density variation may change the microstructure and CPO. The CPO is maintained, the grain shape fabric is reinforced, and the final grain size is larger than the initial one. The preservation of the CPO suggests that most new "dislocation free" nucleii were already close to M1 or M2 orientations before annealing. This suggests bulging as the dominant mechanism for nucleation at the conditions of our experiments (prior to annealing), as it tends to create grains with a closer orientation to the parent grains. Furthermore we report an offset angle between the M1 and M2 submaxima 17.4° greater than what would be expected for a perfect symmetry to the finite extension direction. Other experiments, which did not undergo annealing show a difference in angle of only 1-3° with the perfect symetry. In the TGI0.71 annealed sample, M2 is much closer to M1, as would be expected for a higher finite shear strain. This could be interpreted as a sign of preferential growth of bulging nucleii with orientation closer to M1 than the bulk CPO of the sample before annealing. Conservation of the CPO with annealing also means that if stress is somehow lowered in natural context, and annealing takes place, the CPO could still be interpreted, even if information about grain shape fabric is lost. The conservation of the CPO during annealing has also been observed in experimental and natural samples of quartzite (Heilbronner and Tullis, 2002; Augenstein and Burg, 2011), and rock analog materials such as chloropropane or norcamphor (Park et al., 2001). The reinforcement of the grain shape fabric with elongation toward the finite extension direction (Fig. 2.c) also suggests some oriented grain growth. It is possible that during strain, the formation of a large number of isometric nucleii weakens the shape fabric, but that this evolution is counteracted by grain growth during static annealing.

The full lattice preferred orientations derived from EBSD data reveal that both $<10\bar{1}0>$ and $<11\bar{2}0>$ axes display preferential orientation within the shear plane toward the finite extension direction, as observed in Qi et al. (2019). From theory the 1/3 $<11\bar{2}0>$(0001) slip system is supposed to be dominant in the basal plane (Hondoh, 2000); this assumption is consistent with the present data since such a CPO may result from similar activation of all three $<11\bar{2}0>$ directions, but could also mean an equivalent activation of glide in either $<10\bar{1}0>$ or $<11\bar{2}0>$ directions.

From the WBV analysis of the EBSD maps, we were able to map GNDs and obtain information about their Burgers vectors components. If the density of GNDs increases with shear strain, the proportion of GNDs with Burgers Vector with a

[c]-component seems to be independent of the amount of deformation. However there is a clear increase of the amount of GNDs with a [c]-component Burgers vector in the vicinity of the grain boundaries, as illustrated on Fig. 7. The observation of an increase in proportion close to GBs of the [c]-component GNDs, predicted to be unfavorable compared to basal plane dislocations, is consistent with stress concentrations particularly important close to GBs. As pointed out by Chauve et al. (2017a),

there is a clear connection between the nucleation by bulging mechanism and GNDs located near serrated grain boundaries. The closing of bulges by the formation of a sub-grain boundary was suggested early as important nucleation process in Urai et al. (1986). For instance Chauve et al. (2017b) have shown observations of GND loops in ice that require [c]-component dislocation walls to be closed. This would support the hypothesis that the increase of [c]-component GNDs near grain boundary helps to form GND loops that give birth to new grains. These observations, combined with our present WBV statistical analysis, point towards a possible way to estimate the maximum size of bulging nucleus, which would be lower than 100 $\mu m$ in the present case. It would be important in future studies to test if this critical dimension would vary with temperature or stress, as well as to obtain misorientation data at higher spatial resolution to better constrain the evolution of this higher these zones with strong [c]-component concentration. The fact that the estimated maximum size of bulging nucleus is constant at all finite shear strains suggests that bulging is very active at all creep regime, further stressing its importance for understanding the rheology of ice and other materials at high temperatures.

The pioneering work on 2D modeling of polycrystalline aggregates under simple shear performed by Etchecopar (1977) was able to reproduce CPO with sub-maxima M1 and M2. This was simply done by considering a single slip system (basal slip system for ice) and adding an accommodation process which allows cells to subdivide ( Etchecopar defines this as grain breaking but grain polygonization would have the same kinematic effect) and undergo rigid body rotation. The very good agreement of this simplistic model with evolution of CPOs observed experimentally for ice under shear was emphasized by Bouchez and Duval (1982), who hypothesized that the polygonization processes in ice would be formation of GNDs and kink-bands. In our results few kink bands were observed, but the prevalence of GNDs at most finite shear strains suggests that Bouchez and Duval (1982) supposition is reasonable. Although Etchecopar (1977) modeling is too simplistic to pretend reproducing every shear-induced CPOs in ice, it was useful to raise the likely role of polygonization as an efficient accommodation mechanism for solving strain incompatibility problems.

Modeling of shear in ice has been done by mean-field approaches as in (Castelnau et al., 1996) or more recently by full-field modeling as in (Llorens et al., 2016). Both works reproduced the formation of a strong single maximum CPO from shear strain of about 0.4 and above. Nevertheless, neither the orientation of this single maximum normal to the shear plane, nor the existence of two submaxima as observed at lower strains in the field or experimentally are correctly reproduced. The fact that the single submaximum prescribed is inclined from the tangent to the shear plane is significant, and stands from the fact that these homogenization techniques require the activation of non-basal slip systems. The activation of secondary slip systems, whose contribution to strain has never been proven experimentally, induces a geometrical rotation of the crystal, that is responsible for the modeled inclination of the clustered CPO compared to the vertical. The activity of these secondary slip systems relative to the basal ones is controlled by a parameter that is arbitrarily defined (it has been defined in comparison to experimental observations in Castelnau et al. (1997), using the mean-field VPSC approach, and values different than the one

chosen is the previously cited studies were obtained). The higher the non-basal activity, the softer the mechanical response of the crystal to accommodate the imposed conditions. The geometrical constraint of crystal rotation under shear, owing to the activity of non-basal slip systems, can be artificially relaxed, such as in Wenk and Tomé (1999), by forcing the growth of selected grains, or as in Signorelli and Tommasi (2015), by an association of polygonization and local (within a grain)

relaxation of the strain compatibility constraints.

By comparing these various modeling approaches, and their inclusion of recrystallization mechanisms, it appears that accommodation mechanisms, other than non-basal slip systems, must come into play to explain recrystallization induced shear CPOs in ice. Although we consider that fast grain boundary migration might be an efficient strain accommodation mechanisms, we suggest here that an efficient additional contribution to the CPO reorientation, at the high homologous temperatures of our

experimental studies (and the ones of Bouchez and Duval (1982) or Qi et al. (2017, 2019)), might well be nucleation assisted by polygonisation (or sub-grain boundary rotation).

The analysis of the CPO evolution in the experiments suggests that plastic strain through dislocation creep is probably as commonly claimed, largely dominated by basal slip. Nevertheless, stored GNDs (including non-basal GNDs with [c]-component Burgers vector) allow recrystallization accommodation mechanisms such as nucleation by bulging. The fact that

the PARIS factor initially increases then decreases above $\gamma_{max} =0.42$ enables to make further assumptions about active recrystallization mechanisms and their evolution (or sequence) with increasing strain. Until a strain close to $\gamma_{max} =0.42$, heterogeneities in dislocation storage close to grain boundaries favor spatially irregular grain boundary migration to dominate, creating numerous serrated grain boundaries, as indicated by the increasing PARIS factor. With increasing finite strain, the influence of nucleation gradually increases, forming more isodiametric grains, lowering the PARIS factor, and eventually eras-

ing the M2 maximum. Indeed, SGBs required to close bulges may necessitate strong local stress levels (in order to activate [c]-component dislocations, for instance) that are obtained after a given amount of accumulated strain, while grain boundary migration may occurs since the very beginning of the deformation.

Our results also underline the efforts remaining to match experimental observations and mean-field and full-field numerical modeling. The present data claim for an important role of dynamic recrystallization processes, namely bulging nucleation

and grain boundary migration on the CPO evolution in simple shear, and for the possibility of non-basal dislocation slip to participate to SGB related polygonisation.

## 5 Conclusions

By using state of the art analytical techniques (i.e. AITA, EBSD and Weighted Burgers Vector analysis), we were able to characterize the CPO, microstructure, and geometrically necessary dislocation structures in ice deformed under torsion in the

laboratory at an unprecedented resolution. The experiments, performed at high temperature, up to shear strains of 2, favored dynamic recrystallization observed in natural conditions with slower strain rates such as cold glaciers, ice streams, and some deep ice core areas. The present experiments corroborate previous observations: under simple shear, ice develops a two-

maxima c-axis Crystallographic Preferred Orientation (CPO), which evolves rapidly into a single cluster CPO with c-axis perpendicular to the shear plane.

From new high-resolution observations and analyses, we were able to determine that :

- Abundant sub-grain boundary formation, boundary migration and bulging nucleation are the dominant detected recrystallization mechanisms at these conditions.

- Sub-grain boundaries density and GNDs components are characterized as abundant at all observed finite shear strain, even at this high temperature ($0.97 \cdot T_m$).

- At large finite strain ($\gamma_{max} \leq 1.96$) a significant preferred orientation of both <a> and <m> directions toward the shear direction is observed.

- Annealing of shear strained polycrystalline ice preserve the CPO and creates a strong grain shape fabric elongated toward the finite extension direction.

- Around 30% of Geometrically Necessary Dislocations exhibit a significant $[c]$-component, due to strong local stresses arising from the strong strain heterogeneities, even at these high temperature conditions.

- The density of $[c]$-component of GNDs increases close to grain boundaries. This effect may help the closure of bulging grain boundary and allow efficient bulging nucleation.

So far, full-field modeling approaches that simulate plastic deformation by activating non-basal slip systems in ice were unable to fully reproduce the strongly clustered simple shear CPO observed experimentally and in the nature. As was already suggested by Etchecopar (1977), polygonization may play a crucial role to accommodate dislocation glide dominated by basal slip. Our observations suggest that bulging, associated with sub-grain boundaries mechanisms, could be responsible for this and explain the observed strongly clustered CPO at relatively high strain and high temperature. We conclude that non-basal dislocation slip should not be involved in modeling to obtain realistic CPOs in polycrystalline ice and other anisotropic material, but that GNDs with a $[c]$-components should play a major role in strain accommodation and nucleation processes. We hope that these new results will help the modeling community to provide realistic modeling CPO evolution of simple shear in ice, which are capital for a better predictive ability of ice landmasses evolution in the coming decades and centuries.

*Data availability.* The EBSD, AITA and strain data are available at https://issues.pangaea.de/browse/PDI-20395

*Code availability.* MTEX is a free Matlab® toolbox for analyzing and modeling crystallographic CPOs by means of EBSD or pole figure data (Hielscher and Schaeben, 2008; Bachmann et al., 2010, 2011; Mainprice et al., 2014). It is available on http://mtex-toolbox.github.io/ The Weighted Burgers Vetctor (WBV) analysis is a Matlab® toolbox developed by Wheeler et al. (2009).

PolyLX is a free MATLAB® toolbox for quantitative analysis of microstructures, developed by Lexa (2003), and available on https://petrol. natur.cuni.cz/~ondro/oldweb/polylx:home

*Author contributions.* BJ was the leading author of this paper. Sample preparation and torsion experiments were performed in Grenoble by BJ and LG. EBSD measurements were operated in Montpellier by BJ, TC, MM, FB, and LG. Data analysis were conducted by BJ and TC.
5  Interpretation was done by BJ, TC, MM, AT and DM. BJ and MM wrote the paper with input from all authors.

*Competing interests.* Authors declare no competing interests.

*Acknowledgements.* The authors acknowledge the financial support was provided by the French "Agence Nationale de la Recherche", project DREAM, ANR-13-BS09-0001-01, awarded to MM. This work benefited from support from INSIS and INSU institutes of CNRS. Some of the present work was supported by a grant from Labex OSUG@2020 (ANR10LABEX56) and from INP-Grenoble and UJF in the frame
10  of proposal called "Grenoble Innovation Recherche AGIR" (AGI13SMI15). Some support for BJ also came from the NASA Postdoctoral Program fellowship awarded to BJ, by the NASA Solar System Workings Grant 80NSSC17K0775 and by the Icy Worlds node of NASA's Astrobiology Institute (08-NAI5-0021).
The authors thank Paul Duval for insightful scientific discussions and useful advices on the experimental and theoretical aspects, and John Wheeler for sharing the Weighted Burgers Vector code and providing precious advice on it.

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

**Table 1.** Torsion experiments details for each samples. The sample that was annealed is indicated with an asterisk. $M_{max}$ is the calculated torque, $\tau_{max}$ is the maximum calculated shear stress, $\varepsilon_{max}$ is the maximum measured strain, and $\dot{\gamma}_f$ and $\dot{\varepsilon}_f$ are the final measures shear strain rate and strain rate.

| Sample | radius (mm) | height (mm) | $M_{max}(N.m)$ | $\tau_{max}$ (MPa) | $\varepsilon_{max}$ | Duration (h) | $\dot{\gamma}_f$ (s$^{-1}$) | $\dot{\varepsilon}_f$ (s$^{-1}$) |
|--------|-------------|-------------|----------------|--------------------|---------------------|--------------|-----------------------------|----------------------------------|
| TGI0.012 | 19.15 | 56.05 | 5.8 | 0.4 | 0.006 | 23 | $1.6 \cdot 10^{-7}$ | $0.8 \cdot 10^{-7}$ |
| TGI0.2 | 17.70 | 68.50 | 4.9 | 0.5 | 0.1 | 210 | $5.3 \cdot 10^{-7}$ | $2.7 \cdot 10^{-7}$ |
| TGI0.42 | 16.40 | 57.65 | 4.8 | 0.6 | 0.21 | 270 | $1.1 \cdot 10^{-6}$ | $5.3 \cdot 10^{-7}$ |
| TGI1.96 | 16.25 | 52.20 | 4.9 | 0.6 | 0.87 | 430 | $2.1 \cdot 10^{-6}$ | $8.6 \cdot 10^{-7}$ |
| TGI0.71* | 16.10 | 59.9 | 4.0 | 0.5 | 0.35 | 310 | $1.8 \cdot 10^{-6}$ | $8.4 \cdot 10^{-7}$ |

**Table 2.** Grain size and shape analysis results based on AITA measurements. The sample that was annealed is indicated by an asterisk. $n_g$ is the number of grains used for the analyses and $d_g$ the median of the grain size (with associated interquartile range). Standard deviations are indicated in parentheses.

| Sample | $n_g$ | $d_g$ (mm) | PARIS factor (%) |
|---|---|---|---|
| Unstrained | 558 | 0.7(4) | 2.2(1.3) |
| TGI0.012 | 373 | 0.4(2) | 1.9(1.5) |
| TGI0.2 | 1112 | 0.14(17) | 9.6(12.8) |
| TGI0.42 | 679 | 0.4(4) | 11.5(15.2) |
| TGI1.96 | 926 | 0.12(16) | 5.9(8.4) |
| TGI0.71* | 176 | 1.7(2.4) | 4.9(6.3) |