# Peer review of "Recrystallization processes, microstructure and CPO evolution in polycrystalline ice during high temperature simple shear"

_The Cryosphere, 2018_

## Referee Comment (RC1) · Tielke (Referee) · 4 Dec 2018

**General Comments**

The manuscript "microstructure and texture evolution in polycrystalline ice during hot torsion. Impact of intragranular strain and recrystallization processes" by Journaux et al. describes the results of a series of challenging deformation experiments carried out on ice using a torsional deformation geometry. The detailed microstructural analyses carried out in this study provide a significant advance in characterization of the dis-

location processes that are involved in the development of crystallographic preferred orientation in water-ice aggregates. The organization and writing is generally crisp, but there are a few instances where the English could be more clear. A slightly wider discussion of this work in the context of previous studies, especially with respect to grain-size sensitive deformation and rheological behaviour, would potentially place this manuscript in a more meaningful context. Therefore, once the minor issues below are addressed, I can fully recommend this manuscript for publication in The Cryosphere.

Sincerely,

Jacob Tielke

Specific Comments

1) A deeper discussion of grain-size sensitive deformation mechanisms may be appropriate. For example, the different nature of the distribution of GND's near grain boundaries is an interesting and important observation. One interpretation of this observation is that there is a heterogenous distribution in the magnitude and orientation of stress near grain boundaries. In other words, the presence of grain boundaries may enhance deformation, which would lead to a grain-size sensitive rheological behaviour such as described by Goldsby and Kohlstedt (2001). A grain-size sensitive rheological behaviour may operate at the higher strain conditions where weakening (increasing strain rate with time) is observed. Although it is possible that the weakening is entirely due to the alignment of grains with favourable orientation to operate easy slip at larger shear strain (i.e. geometric softening).

2) A stress exponent of 3 is used to estimate the stress at the outside radius of the samples. Although probably appropriate for this study, it would be better to have a stronger justification for this selection. For example, if a grain-size sensitive mechanism is operation at higher strain (smaller grain size) conditions, a lower value of stress exponent may be more appropriate. Additionally, values of n of 4 have been observed in ice deforming by dislocation creep (e.g., Durham et al., 1983). A stronger justification
for using n=3 would place more confidence in your calculated values of stress and make future comparison of this work more straightforward.

3) A discussion of some more of the recent work on the controls of ice CPO at low strain conditions would help strengthen some of the arguments in this manuscript. For example, Qi et al. (2017, JGR), discussed the importance of stress on controlling the nature of CPO in ice, and noted the importance of grain boundary migration at low stress and lattice rotation at high stress. Although those experiments were carried out at different conditions and using a different deformation geometry, they may provide some insight into the various conditions at which the texture with the M2 maxima are important.

4) Sample TGI0.012 was deformed to very low strain and its data are missing from Figure 1. How certain are you that this small amount of plastic strain was imposed on the sample? The grain size of TSGI0.012 appears smaller than the unstrained sample, which suggests at least some plastic deformation occurred. Was a correction made for the compliance of the rig? Was there any evidence of elastic strain?

Technical Corrections

Title: It is rather unusual to have a two-sentence title, can it be reduced to one sentence?

Page 2, line 3: "compression and extension are the dominant deformation mechanisms" – this is a bit awkward as most people discussion deformation mechanisms as related to flow behaviour, e.g., dislocation creep, diffusion creep, etc. It may be more clear to replace "deformation mechanisms" with "deformation geometries".

Line 6: remove "s" from "orientations"

Line 10: remove "olivine"

Line 23: replace "in return" with "consequentially"

Line 27: add "rate" after "strain" ?

Lines 28 – Page 3, line 2 – The statements made in this paragraph are a bit debatable. I am not sure if ice is a great analogue for mantle rocks. Hundreds (if not thousands) of high temperature experiments have been carried out to study the flow behaviour and microstructural characteristics of mantle rocks. Ice is significantly more anisotropic, in a viscous sense, than olivine and other mantle material.

Page 3: line 10: remove "has"

Line 28: add "either" before "not"

Line 29: remove space after "mechanisms"

Page 4 Line 6: replace "packing evenly" with "evenly packing"

Line 17: replace "control visually" with "allow for observation of"

Page 5 Line 18: replace "does" with "do"

Line 31: replace "identify" with "identified"

Page 6 Line 20: replace "reminded" with "noted"

Line 1: here you say the noise was too large to distinguish any primary creep hardening in TGI0.012

Line1: add space after at end of sentence

Line 5-6: Can you say for certain the TGI0.012 stayed in the primary creep regime? The strain was very small and the data for that experiment is not presented in Figure 1

Line 21: add "essentially" before "random"

Line26-27: Did TGI0.012 really achieve the strain indicated? How did you account for compliance and elastic deformation of the sample?

Interactive
comment

Figure 1: missing data for TGI0.012

Figure 1 caption: replace "experiments" with "experiments", or change this sentence unless you add data for TGI0.012 to the figure. Replace "The blank part of the curve corresponds to" with "the blank parts of some of the curves correspond to". Remove "represented".

Line 10: add "(by gamma = 0.2)

Figure 3: This is a nice plot but the large gab between gamma 1 and 2 suggests that maybe it is worthwhile mapping an axial section of TGI1.96? This would allow you to calculate the J-index at all strains by mapping form the center (almost unstrained) to the outside (almost gamma=2) of the sample.

Line 1: replace "develops" with "develop"

Line 9: remove sentence about spatial resolution, that is already in the methods section

Figure 5: what is represented by the pole figures? c-axes?

Line 2: add "dislocations" after "those"

Line 9: add "investigated" after "strain"

Line 3: replace "identify" with "identified"

Line 5: replace "kinds" with "types"

[Figure]

Line 16: remove " t "

Line 15: replace "these study" with "those studies"

Line 7: Replace "advices" with "advice"

---

## Referee Comment (RC2) · Prior (Referee) · 11 Dec 2018

Review of **Journaux, Baptiste et al:** "Microstructure and texture evolution in polycrystalline ice during hot torsion. Impact of intragranular strain and recrystallization processes."

By **Dave Prior,** University of Otago. 9th December 2018.

**General Comments:**

This is an excellent contribution that presents new data on the evolution of microstructure and crystallographic preferred orientation (CPOs) of polycrystalline ice during simple shear up to a shear strain of ~2. The paper is reasonably well-written and is well-illustrated. I enjoyed reading it. The wmv analysis is particularly nice and provides a good analytical template for other researchers (including me!). I have some significant scientific discussion points that I would like the authors to consider and I have some suggested modifications. In addition to the comments in this document, I have annotated a pdf of the paper. I hope that my comments are useful.

There is significant complementarity between this paper and a paper that we also have in Cryosphere Discussions ([https://www.the-cryosphere-discuss.net/tc-2018-140/](https://www.the-cryosphere-discuss.net/tc-2018-140/)), with Chao Qi as first author. I will refer to this paper as (Qi et al., 2018) in this review. I hope that the authors can relate some of their observations to the ones that we have made; we will endeavour to do the same.

**Scientific Discussion:**

1. Some of the description of the CPOS is misleading or incorrect.
   a. The **<11-20> and <10-10>** in the high strain sample ($\gamma$=1.96) are not randomly distributed within the girdle. The <11-20> and <10-10> both have broad maxima, parallel to the shear direction, of ~ 4x m.u.d. and ~3 x m.u.d. respectively. These compare to minima within the girdle of ~ 2x m.u.d.

[Figure]

This level of <a> and <m> alignment is comparable to that shown for the highest shear strain data at -5C in fig 4 of (Qi et al., 2018). Additionally the ratio to the <c> axis maximum (max <a> ~ max (c)/y where y is between 2 and 4) is very similar to the highest shear strain data at -5C and all data at -20C and -30C in fig 4 of (Qi et al., 2018). The alignment of <a> and <m> orientations is important. This might provide a cool tool for assessing shear directions in the analysis of naturally deformed ice so it needs to be documented. <a> and <m> being co-aligned matches our data and is intriguing. At present I do not have a coherent explanation for this. I'd be interested to hear your views on this.

b. You have not commented on the **shape of the M1 and M2** maxima. In virtually all experimentally sheared polycrystalline ice samples these maxima are elongated in a direction perpendicular to the shear direction (see discussion in (Qi et al., 2018) and in our response to a Maurine Montagnat comment on this in the discussion section). Sometimes the elongated maxima (both M1 and M2) are actually each double maxima, with the profile plane as a mirror plane. The vast majority of naturally sheared ice samples do not have elongated maxima, the contours of the maxima match small circle distributions (e.g. (Hudleston, 1977)). This point of difference between experiment and nature is important and as such it is important that the shape of the M1 and M2 maxima from experiments is described.

[Figure]

The high strain ($\gamma$=1.96) M1 is clearly elongated in the direction perpendicular to shear. I have superposed small circles, with their cone axes on the primitive, on the figure above to emphasise this point. M1 in the lower strain experiments is not so clearly elongated. In the annealed experiment the contours match the small circles, and it looks like this is the case for the lower strain experiments. In our experiments (Qi et al., 2018) elongation increases with shear strain.

M2 in the $\gamma$=0.42 experiment is elongated, with a double maximum (labeled above max1, max2), with the profile plane as a mirror plane. The $\gamma$=0.42 experiment may also show this but I can't tell from the figure. Interestingly M2 in the annealed sample does not look elongated. This could be an important point. Does annealing remove the cluster elongation? One of the reasons we adopted a different reference frame in (Qi et al., 2018), with the pole to shear plane in the middle of the stereonet, is that it makes it easier to see cluster shapes, as shown below in a re-analysis of the (Bouchez and Duval, 1982) data. The highest and lowest strain samples in these data have elongated M1, the medium strain sample does not.

Replotting of Bouchez and Duval 1982 data. All processing in MTEX

c. I think you need to be a little more precise in description of the symmetry of the M1, M2 maxima pair with respect to the finite elongation direction. I think this is a cool observation and potentially of some value, but the symmetry is far from perfect. Below I have plotted up some traces for M1 and M2 (red lines), with angles measured from the top of the stereonet. The green line has equal angles to the two red traces. Superficially this green line is close to the finite extension direction (ED), but if I plot the expected M2 trace (yellow line) assuming it has the same angle to ED as M1 (and adjusting ED for for M1 not being at 0 degrees in the two lowest strains) then the observed M2 is anticlockwise of the yellow line for the three lowest strain, most markedly for the annealed sample. The symmetry you describe is approximate.

[Figure]

Another way of looking at this is to plot the angle between M1 and M2 against shear strain. Below is a modified version of fig 8 from (Qi et al., 2018) with the addition of your data (big red dots) and a line (pink) that predicts the position of M2 if it has the same angle to the finite extension direction as M1. This is quite an interesting addition to the plot as very broadly the red data points (high T experiments: not just yours) do follow the path of the pink line, but at slightly lower angles? Is M2 at high T and low shear strain (<=~2) related to the orientation of the finite strain ellipsoid?

[Figure]

2.  The **description/ documentation of the experimental set up** needs to be improved.  Please provide some key diagrams that show the experimental set up.  Torsion is an important deformation kinematic and the torsion experiments you show here and the classic work of (Bouchez and Duval, 1982) represent significant contributions to our understanding of ice with direct application to polar ice sheets and glaciers. I believe that torsion is an important defomation kinematic to explore more fully in the future. The picture in (Duval, 1976) and the words in (Bouchez and Duval, 1982), (Duval, 1976) and presented here are insufficient for someone to reproduce the experimental set up. It would be great if you could present (maybe in supplementary information) some diagrams that show the mechanics of the deformation apparatus. There is one particular aspect that I think is of paramount importance. I think that this apparatus is constrained to deliver simple shear, with no shortening or extension normal to the shear plane.  If this is the case I presume that the "platens", that deliver the torque, are fixed so that they cannot move normal to the shear plane. This is important so that we can be clear which experiments are simple shear only, and which comprise simple shear with a component of shortening (or extension). This is not necessarily the same as having zero normal stress on the shear plane. (Li et al., 2000) (a key paper that is not cited in your work) point out that direct shear experiments using a "Jacka" rig, with the normal load set as zero still experience shortening/ extension normal to the shear plane (and that the magnitude depends on sample geometry). Furthermore they suggest that an experiment with fixed platens will generate shear plane normal stresses of 0.1 to 0.2MPa. In my view a constrained (by fixed platens) simple shear

experiment is great - it's a clear kinematic end member. We do need to be absolutely clear about the experimental kinematics and the implications the kinematics have for stress, rheology and microstructure. What are the kinematics and dynamics of naturally deforming ice systems is yet another matter. I can imagine some scenarios (e.g. ice stream margins) where perfect simple shear may occur and others (e.g. basal zones) where shear with shortening parallel to the shear plane occurs.

3. The **mechanical data** are a bit puzzling. The focus of this paper is the microstructure, and I don't think the questions about the mechanical data affects substantially the microstructural observations and interpretations, but I would like to see a bit more analysis of the data. The key problem for me is that the applied shear stress should be the dominant control of the shear strain rate (whether secondary, tertiary of at a ~ given strain in transient creep), given that your temperature and starting materials were nominally the same for all experiments. A shear stress of 0.6MPa vs 0.50.5MPa should give a ~ doubling of strain rate (for n between 3 and 4). The secondary creep rate for TG10.42 (0.6MPa) is slower than that for TG10.71 (0.5MPa) and faster than TG10.2 (also 0.5MPa). In the text this is attributed to "variability of grain size and textures". This could be true, but in needs to be unpicked in a bit more detail.

   The method used to fabricate the starting material sounds the same as that we use (except that we do not anneal) as described in (Stern et al., 1997). We have looked at >10 samples of starting materials made by the same methods in four different labs (Otago, MIT, UPenn, UCL) and all have very very similar grain size distributions, mean grain size and random CPO; an example is in fig 1a in (Qi et al., 2017). I cannot see that the annealing will affect the CPO and annealing at consistent T and time should give the same grain size distribution. Do you have initial g size data from more than one sample? We can estimate what grain size differences would be needed to explain the variations in secondary creep rate. The ratio of secondary creep rates of the two samples deformed at 0.5MPa is about 2 (estimated from slopes on fig: would be good to provide an enlargement of secondary creep region, as you have done for primary creep region).

[Figure]

   Using the grain size exponent (-1.4) from (Goldsby, 2006; Goldsby and Kohlstedt, 2001)jb this would require the relative mean grain sizes of the two samples to be ~ 1.7. (e.g 1.5mm and 0.9mm). This grain size exponent may be a bit large. A more conservative estimate (related to similar starting materials) comes from using the peak stress (= secondary creep) data in (Qi

et al., 2017), fig 3. This gives an ∼ grain size exponent of -0.8, requiring a grain size ratio of ∼2.3 (e.g 1.5mm and 0.65mm) to explain the strain rate differences at 0.5MPa. I am pretty sure that your original grain sizes do not vary by a factor of ∼2, so grain size is unlikely to provide an explanation for the variability in mechanical data.

Although it seems likely that your bulk CPO is random in all starting materials, it is worth considering whether the sample cross section contains enough grains to give the mechanical properties of a random CPO. This was clearly an issue for us deforming 1 inch diameter samples with a ∼5mm grain size (Craw et al., 2018): in this case a cross section may contain only 10 or 20 grains and the peak stress (= secondary minimum) data do not have a systematic relationship to strain rate. In your case there should be ∼ 500 grains in a 35mm diameter cross-section so I would have thought this effect is unlikely to be significant.

It seems unlikely to me that the variations in strain rate relate to variability in the starting material. In this case it's worth looking back at the experimental set up. How is stress transferred from the rotational drive platens (this needs describing- see point 2) to the sample? Is there a possibility that there is some slippage (frictional loss) or other parameter that varies from one sample to the next so that the torque is not all transferred to shear stress on the sample?

4. The **discussion of modeling** is rather black and white and superficial. Numerical models and physical experiments all have limiting boundary conditions. All models and experiments show us something and none match nature, primarily because we cannot access natural conditions and have uncertainties about natural boundary conditions. Linking physical experiments to numerical models is important as we have much more control on the boundary conditions in both cases: so we learn more about our understanding of processes. However the crucial thing for both experiments and models is that we are clear about what we learn from them. I think having a model that is able to simulate fully CPO and microstructure evolution at high strains is still a way off.  All steps on the way to achieving this are valuable and a discussion that implicates that one model is right and another wrong is inappropriate: in demeans what we learn from the models.

I agree that the  Etchecopar model as used in (Bouchez and Duval, 1982) matches quite well your data and most of the "hot" shear data (see the red symbols in M1-M2 angle vs shear strain graph posted earlier: Etchecopar model also plotted on this as hollow black squares). The problem is that these are the only data it fits, so if this model is applicable it tells us only part of the story. The model does not predict the drop off to single maxima by shear strain of 2 in (Li et al., 2000); maybe this is a kinematic difference between simple shear and simple shear plus some strain normal to shear. The model does not match the minimal "colder" data we have, most particularly the -30 data from (Qi et al., 2018). The FFT model (Lebensohn, 2001) gives a remarkable match to experimental observations of intragranular deformation at low strain (Grennerat et al., 2012; Lebensohn et al., 2009). This is the code used to simulate shear deformation in the models by (Llorens et al., 2016; Llorens et al., 2017). The fact that the same model works well at low strain and less well at high strain tells us something. The bulk CPOs in

Lloren's models do not have double maxima, but the double maxima are there when only the high strain rate data are used (see Llorens, 2017 fig 5i) and the angle between maxima in the deformation only models evolves in a way that matches the -30 experimental data we have (Qi et al., 2018). Addition of recrystallization into the model changes the result, although not in a way that gives a really clear match to observations. There is no real conclusion here apart from this: both models and experiments are important. Probably most important is to design experiments that enable clear boundary condition matches to numerical models. That is the really beautiful thing about the columnar ice work at low strain e.g. (Grennerat et al., 2012). At high strain and in shear matching of model and experimental boundary conditions is rather harder.

**Clarity of writing**

The bulk of the text is well-written. The clarity of the writing is not as good in the discussion and not good at all in the conclusions.

The discussion would benefit from some shortening and restructuring. The discussion starts with a reminder of the key observational data and I think it would be very helpful to the reader if you added a schematic diagram to highlight these key observations. This would then give a clear framework for ongoing discussion.

The conclusions needs to have clear statements on what are the new factual observations and what are the interpretations of those observations.

The abstract should be a concise summary of the new findings and some short statement about importance. The abstract contains an extended statement of background that is better placed in the introduction (it is in fact already in the introduction).

I would go for a simpler title: "Evolution in polycrystalline ice microstructure during progressive high temperature shear" ????

**Technical/ terminological/ picky things (in no particular order):**

5. It would be great if you could show full grain size distributions (frequency plots). You are correct that the mean is not a great scalar to represent recrystallized grain size statistics. Grain size distributions could be represented as an extra row in figs 2 and 4 (it would be nice to compare the AITA and EBSD measures- I don't expect them to be the same: see (Cross et al., 2017))
6. Please put the number of grains that correspond to each pole figure on figs 2 and 4 or in a table. This is important in comparing data sets.
7. If you can, show point stereonets as well as contoured nets. The contoureing hides a lot of information.
8. The statement on page 2, line 26 states that the "texture can increase shear strain rate (word "rate" missing) by a factor of … ". There is a clear correlative relationship of weakening and CPO but a causative relationship is not established. Weakening in ice from secondary to tertiary creep correlates with development of a CPO. It is intuitive that the CPO developed in shear

facilitates further shear. However similar weakening occurs in cold axial shortening where the CPO (cluster of c-axes parallel to shortening) would intuitively make further axial shortening harder e.g. -30 experiments right hand column of fig 3 in (Craw et al., 2018), mechanical data in fig 10. Other changes correlate with weakening, most particularly grain size changes (as documented in your paper and elsewhere). In the geological literature grain size reduction is often thought of as the main cause of weakening. In reality CPO, grain size and other microstructural parameters all change in correlation to change in mechanical behavior. It is unlikely that the mechanical evolution is caused by changes to just one of these sample parameters.

9.  I don't think that Kamb's idea that CPO is independent of of T, strain rate or stress is confirmed (P3, L11). The data in (Qi et al., 2018) show that in shear the CPO changes with T. (Qi et al., 2017) show that in axial shortening CPO is sensitive to stress or strain rate (the two cannot be separated). It is reasonable that the stress/ rate effect will also apply in shear. Using Huddleston's data in comparison to experiments is complex as both T and rate change. The lower rate has a similar effect to deforming hotter.

10. The statement on page 5, line 8 is incorrect. Cryo EBSD of ice is not (in general) limited to samples of ~10 by 20mm. In terms of published data there is a map in (Prior et al., 2015) (fig 12) of 80 by 30mm, the data in (Wongpan et al., 2018) has maps up to 40 by 40mm etc. Most of the CPO data we publish from experimental samples come from 25.4 by 40mm samples, our shear data CPOs in (Qi et al., 2018) are from elliptical shear surfaces of 25 by ~ 30mm. For natural samples we routinely work on samples of ~60 by 40mm and with suitable cold stage modifications I don't see why 100 by 50mm is not achievable. EBSD maps with the same dimensions as your AITA maps are possible now. If the Montpelier machine has a sample size limitation and this limitation is important to the paper, then link the limitation to that instrument, otherwise just delete the statement about size limitation. I guess if it the Montpelier machine does have a limitation it must be to do with cold stage tethering (gas pipes) or camera position limiting WD, as the sub-stage is designed for very large stages/samples (Seward et al., 2002).

11. Please provide enough information for the reader to understand how surface sublimation is managed. What I mean by this is; how is frost removed from the sample. There will be a frost layer on the sample surface as it goes into the SEM that would prevent EBSD (needs only ~ 10-20nm to do this). The two main ways of removing the frost are to heat the stage (Iliescu et al., 2004; Weikusat et al., 2011) or to cycle through pressure (Prior et al., 2015). I recall Andrea Tommasi telling me that the sample is just put in the SEM and it works. In this case I infer that the sublimation to remove the frost occurs on the down pressure cycle and that the sample is warm enough when put in the SEM to give a path through PT space where the sample goes into the vapour field (see fig7 in (Prior et al., 2015). In this case it would be useful to know the sample temperature on insertion and the pressure sequence: do you go to high vacuum then to controlled gas pressure or directly to controlled gas pressure?

12. Please say in figure captions if pole figures are equal area or equal angle. I think they are equal area from the shapes of maxima (the projection affects shape analysis of maxima).
13. It would be really cool to see a radial section of the sample: to see how microstructure changes with strain in a single sample (e.g. see (King et al., 2011). I'm not suggesting this is needed for this paper- just something cool to do.
14. There are a few key references on experimental shear of ice that are missing and should be cited. These include (Budd et al., 2013; Li et al., 2000; Wilson and Peternell, 2012).
15. There are several published papers that show a lack of CPO change in rocks during annealing. Some of these should be cited.(Augenstein and Burg, 2011; Heilbronner and Tullis, 2002; Ree and Park, 1997). I know there are others in calcite and olivine but can't find them just now.
16. Throughout this paper the term "texture" is used with the meaning common in metallurgy and materials science. There is a very small community of geoscientists who use "texture" in this way and no glaciologists that I know of. For the vast majority of the geoscience community "texture" means the spatial relationships of phases and grains and their internal structures. To most geoscientists, texture is what you would see down a microscope (in a petrographic examination for example) and is broadly synonymous with the term microstructure. The terms "crystallographic preferred orientation" (CPO: which you use in the intro) or "lattice preferred orientation" (LPO) are much better as they are explicit. If you want this paper to have wider readership/ uptake, remove the word texture throughout and replace with CPO. It is also worth (in the intro) relating this terminology to the word "fabric" and/or the acronym "COF" (crystal orientation fabric) as commonly used in glaciology. I avoid using the term fabric (except in explanations of how terminology matches up) as metallurgists use this term to mean microstructure.
17. It is not really clear what are the observations you use to constrain the dimensions of the bulging nucleus.
18. I don't follow the discussion related to nucleation in the section where the annealing is discussed. Grain size increases during the annealing so nucleation is unnecessary. If you are talking about relationships that might be relevant to nucleation prior to the annealing then this needs to be made clear.
19. Bulges cut off by rotation of a subgrain boundary was first suggested (described from see through experiments) by Janos Urai (I think). You should reference (Urai et al., 1986).
20. Spontaneous (random) nucleation? I have a problem with this - it is a bit of magic with no physically realistic explanation.
21. The conditions of your experiments are not close to those in cold glaciers and ice streams (page 18, line 5). Your **slowest** transient strain rate is 2.7E-7s$^{-1}$ which corresponds to a 100m thick shear zone having a velocity difference across it of 850m/yr. The tertiary strain rate in your high strain experiment corresponds to ~2700m/yr difference across a 100m shear zone. I'm not so familiar with temperate glaciers but such shear rates do not exist in polar ice sheet systems eg (Bons et al., 2018; Rignot et al., 2011). Even fast ice stream shear margins max out below 1E-9s$^{-1}$ (Bindschadler et al., 1996; Jackson,

1999; Jackson and Kamb, 1997). The strain rate has a significant effect on the microstructure and the CPO (Hirth and Tullis, 1992; Qi et al., 2017; Tullis, 1972): increasing strain rate has a comparable effect to decreasing temperature. It is not possible to do an experiment to significant strain at natural conditions. Instead experiments need to provide scaling relationships that allow us to predict the effects of T, strain rate (stress) etc on rheology and CPO/microstructure (with the complication that there are feebacks where CPO/microstructure affect the rheology).

END

**References**

Augenstein, C., and Burg, J. P., 2011, Natural annealing of dynamically recrystallised quartzite fabrics: Example from the Cevennes, SE French Massif Central: Journal Of Structural Geology, v. 33, no. 3, p. 244-254.

Bindschadler, R., Vornberger, P., Blankenship, D., Scambos, T., and Jacobel, R., 1996, Surface velocity and mass balance of Ice Streams D and E, West Antarctica: Journal of Glaciology, v. 42, no. 142, p. 461-475.

Bons, P. D., Kleiner, T., Llorens, M. G., Prior, D. J., Sachau, T., Weikusat, I., and Jansen, D., 2018, Greenland Ice Sheet – Higher non-linearity of ice flow significantly reduces expected basal motion: Geophysical Research Letters, v. Early View, no. doi:10.1029/2018GL078356.

Bouchez, J. L., and Duval, P., 1982, The fabric of polycrystalline ice deformed in simple shear - experiments in torsion, natural deformation and geometrical interpretation: Textures and Microstructures, v. 5, no. 3, p. 171-190.

Budd, W. F., Warner, R. C., Jacka, T. H., Li, J., and Treverrow, A., 2013, Ice flow relations for stress and strain-rate components from combined shear and compression laboratory experiments: Journal of Glaciology, v. 59, no. 214, p. 374-392.

Craw, L., Qi, C., Prior, D. J., Goldsby, D. L., and Kim, D., 2018, Mechanics and microstructure of deformed natural anisotropic ice: Journal of Structural Geology, v. 115, p. 152-166.

Cross, A. J., Prior, D. J., Stipp, M., and Kidder, S., 2017, The recrystallized grain size piezometer for quartz: An EBSD-based calibration: Geophysical Research Letters, v. 44, no. 13, p. 6667-6674.

Duval, P., 1976, Temporary or permanent creep laws of polycrystalline ice for different stress conditions: Annales De Geophysique, v. 32, no. 4, p. 335-350.

Goldsby, D. L., 2006, Superplastic Flow of Ice Relevant to Glacier and Ice-Sheet Mechanics, in Knight, P. G., ed., Glacier Science and Environmental Change: Oxford, Blackwell, p. 308-314.

Goldsby, D. L., and Kohlstedt, D. L., 2001, Superplastic deformation of ice: Experimental observations: Journal of Geophysical Research-Solid Earth, v. 106, no. B6, p. 11017-11030.

Grennerat, F., Montagnat, M., Castelnau, O., Vacher, P., Moulinec, H., Suquet, P., and Duval, P., 2012, Experimental characterization of the intragranular strain field in columnar ice during transient creep: Acta Materialia, v. 60, no. 8, p. 3655-3666.

Heilbronner, R., and Tullis, J., 2002, The effect of static annealing on microstructures and crystallographic preferred orientations of quartzites experimentally deformed in axial compression and shear, in DeMeer, S., Drury, M. R., DeBresser, J. H. P., and Pennock, G. M., eds., Deformation Mechanisms, Rheology and Tectonics: Current Status and Future Perspectives, Volume 200, p. 191-218.

Hirth, G., and Tullis, J., 1992, Dislocation Creep Regimes in Quartz Aggregates: Journal of Structural Geology, v. 14, no. 2, p. 145-159.

Hudleston, P. J., 1977, Progressive Deformation and Development of Fabric Across Zones of Shear in Glacial Ice, in Saxena, S. K., Bhattacharji, S., Annersten, H., and Stephansson, O., eds., Energetics of Geological Processes: Hans Ramberg on his 60th birthday: Berlin, Heidelberg, Springer Berlin Heidelberg, p. 121-150.

Iliescu, D., Baker, I., and Chang, H., 2004, Determining the orientations of ice crystals using electron backscatter patterns: Microscopy Research and Technique, v. 63, no. 4, p. 183-187.

Jackson, M., 1999, Dynamics of the shear margin of ice stream B, West Antarctica [PhD: Caltech, 118 p.

Jackson, M., and Kamb, B., 1997, The marginal shear stress of Ice Stream B, West Antarctica: Journal of Glaciology, v. 43, no. 145, p. 415-426.

King, D. S. H., Holtzman, B. K., and Kohlstedt, D. L., 2011, An experimental investigation of the interactions between reaction-driven and stress-driven melt segregation: 1. Application to mantle melt extraction: Geochemistry Geophysics Geosystems, v. 12.

Lebensohn, R. A., 2001, N-site modeling of a 3D viscoplastic polycrystal using Fast Fourier Transform: Acta Materialia, v. 49, no. 14, p. 2723-2737.

Lebensohn, R. A., Montagnat, M., Mansuy, P., Duval, P., Meysonnier, J., and Philip, A., 2009, Modeling viscoplastic behavior and heterogeneous intracrystalline deformation of columnar ice polycrystals: Acta Materialia, v. 57, no. 5, p. 1405-1415.

Li, J., Jacka, T. H., and Budd, W. F., 2000, Strong single-maximum crystal fabrics developed in ice undergoing shear with unconstrained normal deformation, *in* Hutter, K., ed., Annals of Glaciology, Vol 30, 2000, Volume 30, p. 88-92.

Llorens, M. G., Griera, A., Bons, P. D., Lebensohn, R., evans, L., Jansen, D., and Weikusat, I., 2016, Full-field predictions of ice dynamic recrystallisation under simple shear conditions: Earth and Planetary Science Letters, v. 450, p. 233-242.

Llorens, M. G., Griera, A., Steinbach, F., Bons, P. D., Gomez-Rivas, E., Jansen, D., Roessiger, J., Lebensohn, R. A., and Weikusat, I., 2017, Dynamic recrystallization during deformation of polycrystalline ice: insights from numerical simulations: Philosophical Transactions of the Royal Society a-Mathematical Physical and Engineering Sciences, v. 375, no. 2086.

Prior, D. J., Lilly, K., Seidemann, M., Vaughan, M., Becroft, L., Easingwood, R., Diebold, S., Obbard, R., Daghlian, C., Baker, I., Caswell, T., Golding, N., Goldsby, D., Durham, W. B., Piazolo, S., and Wilson, C. J. L., 2015, Making EBSD on water ice routine: Journal of Microscopy, v. 259, no. 3, p. 237-256.

Qi, C., Goldsby, D. L., and Prior, D. J., 2017, The down-stress transition from cluster to cone fabrics in experimentally deformed ice: Earth and Planetary Science Letters, v. 471, p. 136-147.

Qi, C., Prior, D. J., Craw, L., Fan, S., Llorens, M. G., Griera, A., Negrini, M., Bons, P. D., and Goldsby, D. L., 2018, Crystallographic preferred orientations of ice deformed in direct-shear experiments at low temperatures: The Cryosphere Discuss.\, v. 2018\, p. 1\-35\.

Ree, J. H., and Park, Y., 1997, Static recovery and recrystallization microstructures in sheared octachloropropane: Journal of Structural Geology, v. 19, no. 12, p. 1521-1526.

Rignot, E., Mouginot, J., and Scheuchl, B., 2011, Ice Flow of the Antarctic Ice Sheet: Science, v. 333, no. 6048, p. 1427-1430.

Seward, G. G. E., Prior, D. J., Wheeler, J., Celotto, S., Halliday, D. J. M., Paden, R. S., and Tye, M. R., 2002, High-temperature electron backscatter diffraction and scanning electron microscopy imaging techniques: In-situ investigations of dynamic processes: Scanning, v. 24, no. 5, p. 232-240.

Stern, L. A., Durham, W. B., and Kirby, S. H., 1997, Grain-size-induced weakening of H2O ices I and II and associated anisotropic recrystallization: Journal Of Geophysical Research-Solid Earth, v. 102, no. B3, p. 5313-5325.

Tullis, J., 1972, Preferred Orientations of Experimentally Deformed Quartzites: Transactions-American Geophysical Union, v. 53, no. 4, p. 514-&.

Urai, J. L., Means, W. D., and Lister, G. S., 1986, Dynamic recrystallization of Minerals, *in* Hobbs, B. E., and Heard, H. C., eds., Mineral and Rock Deformation (Laboratory Studies), Volume 36, p. 161-200.

Weikusat, I., De Winter, D. A. M., Pennock, G. M., Hayles, M., Schneijdenberg, C. T. W. M., and Drury, M. R., 2011, Cryogenic EBSD on ice: preserving a stable surface in a low pressure SEM: Journal Of Microscopy-Oxford, v. 242, p. 295-310.

Wilson, C. J. L., and Peternell, M., 2012, Ice deformed in compression and simple shear: control of temperature and initial fabric: Journal of Glaciology, v. 58, no. 207, p. 11-22.

Wongpan, P., Prior, D. J., Langhorne, P. J., Lilly, K., and Smith, I. J., 2018, Using electron backscatter diffraction (EBSD) to measure full crystallographic orientation in Antarctic land-fast sea ice.: Journal of Glaciology, v. 64, 771-780.

---

## Short Comment (SC1) · 7 Jan 2019

We would like to take the opportunity to comment on several statements in the manuscript "Microstructure and texture evolution in polycrystalline ice during hot torsion. Impact of intragranular strain and recrystallization processes" by Baptiste Journaux and co-authors. In this manuscript, the authors report the results of microstructure and texture evolution in polycrystalline ice during torsion experiments.

The manuscript is in general well-structured and the results are innovative. Two reviewers already provided constructive comments and recommendations. However, we consider that some aspects are not correctly addressed in this study, especially when the experimental results obtained are compared with previously published full-field numerical simulations. In different parts of the manuscript the authors state that results of full-field simulations by Llorens et al. (2017) do not match those observed in experiments (e.g. 20-25, page 3; 1-6 page 17; 30-35, page 17). The authors essentially state that numerical simulations are unable to predict the "double sub-maximum of the c-axis preferred orientation or the interlocking grain boundaries" observed in simple shear deformation experiments.

First of all, we would like to note that the settings of the experiments are different from those of the numerical simulations, because the shear strain rates in experiments are between four and five orders of magnitude higher than in the numerical simulations of Llorens et al (2017). Therefore, differences in grain boundary geometries can be expected, as the balance between the increase of system energy due to plastic deformation (e.g. strain stored energy, but also boundary energy) and its reduction due to recrystallisation is strain-rate or time dependent. The much higher system energy in the experiments, compared to numerical simulations, becomes clearly visible in the annealing experiment, in which fast grain boundary migration quickly sets in, together with an increase of grain-size and formation of sharp grain boundaries. These observations are in agreement with results by Llorens et al. (2016b), who explored the influence of shear strain rate and dynamic recrystallisation during simple shear deformation. These simulations clearly reveal the influence of strain rate on the increase of internal misorientation, even though the simulations by Llorens et al. (2016a,b) did not yet include the formation of new high-angle grain boundaries by subgrain rotation (polygonisation) or nucleation. These processes were included in Steinbach et al. (2017) and Llorens et al. (2017). Contrary to the claim by Journaux et al., Steinbach et al. (2017) showed that interlocking grain boundaries and even grain dissection (i.e., grain division due to fast, abnormal grain boundary migration) are reproduced by the numerical simulations, depending on strain rate and settings of dislocation energy and effective mobility of grain

boundaries. The simulations by Steinbach et al. (2017) were carried out in pure-shear, but as revealed by Llorens et al. (2017), the differences between polycrystals deformed in pure- and simple shear conditions are small if the same set-up of deformation and recrystallisation is used.

A second aspect that we would like to clarify is the double-maxima (M1 and M2) of the c-axis in the experiments and simulations. Predictions by Llorens et al. (2017) show that the bulk c-axis maximum is obliquely oriented with respect to the shear plane. However, if only regions with maximum shear strain-rate are taken into account, the double maxima can be observed in orientations similar to those in experiments (Llorens et al., 2017). The evolution of these double maxima with increasing deformation is discussed in the manuscript by Qi et al., (2018; Cryosphere Discussions (https://www.the-cryosphere-discuss.net/tc-2018-140/), as a reviewer has already indicated. The arrangement of the c-axis oblique to the shear plane in the bulk texture is a consequence of strain localisation and the associated formation of low- and high-strain (-rate) domains in the models (Qi et al., 2018). The CPO evolves at a different rate in these domains: the c-axis quickly rotates to become parallel to the shear plane normal in high-strain domains, while it rotates slowly in the low-strain domains (e.g., Llorens et al 2016a,b; 2017). As low-strain regions are volumetrically larger than high-strain domains, the bulk CPO mainly reflects the CPO of low-strain domains. The origin of strain localisation is a consequence of the mechanical anisotropy produced by the difference in flow resistance between non-basal and basal slip systems (parametrised using the parameter A). At least for numerical simulations, an increase of A tends to increase the mechanical anisotropy of the aggregate and favours strain localisation. In contrast, low anisotropy tends to induce more macroscopically homogenous deformation (e.g. Gomez-Rivas et al., 2017) and low-strain domains are volumetrically smaller. This has implications for c-axis orientations: low A values produce a faster rotation of the c-axis normal to the shear plane (and higher activation of non-basal planes) than large A values (resulting in lower activation of non-basal planes).

In a torsion experiment, the torque component between the top and bottom of the sample creates a shear gradient, similar to simple shear deformation. However, a strong limitation of torsion experiments is that the main strain gradient develops across the diameter of the sample, as the centre of the sample remains undeformed, and the maximum deformation is attained at the external surface of the sample. Unfortunately, Journaux et al., only show tangential sections (subparallel to large-strain domains). It would be a very useful piece of information for the reader and community if they authors could also show a perpendicular section that includes the centre of the sample. This section may provide more information than the tangential ones because they allow observing the variation of CPO with strain, and therefore the effect of strain localisation in the sample. Llorens et al. (2017) did show a difference in CPO for high- and low-strain domains, which is discussed in Qi et al. (2018) in terms of the M1 and M2 submaxima. The relative strength of these maxima depends on the amount of shear localisation. This is probably different in the torsion experiments and the simulations. The torsion experiments are likely to reduce localisation in the tangential plane, as developing shear bands need to laterally penetrate into the low bulk-strain parts in the centre of the sample. In the semi-2D numerical simulations, shear localisation is laterally unconstrained, which may explain the small volume fraction of high-strain material.

In conclusion, we disagree with the statement that the results of numerical simulations are in disagreement with the results of the torsion experiments. Clearly, there are differences between the two as both approaches are different and have their own advantages and limitations. We would urge the authors to acknowledge these.

Sincerely,

Maria-Gema Llorens, Universitat Autònoma de Barcelona, mariagema.llorens@uab.cat

Paul Bons, Eberhard Karls University Tübingen, paul.bons@uni-tuebingen.de

Albert Griera, Universitat Autònoma de Barcelona, albert.griera@uab.cat

References

Gomez-Rivas, E., Griera, A., Llorens, M.-G., Bons, P.D., Lebensohn, R.A., Piazolo, S. (2017) Subgrain Rotation Recrystallization During Shearing: Insights From Full-Field Numerical Simulations of Halite Polycrystals. Journal of Geophysical Research: Solid Earth, https://doi.org/10.1002/2017JB014508.

Llorens, M.-G., Griera, A., Steinbach, F., Bons, P.D., Gomez-Rivas, E., Jansen, D., Roessiger, J., Lebensohn, R.A., Weikusat, I. (2017) Dynamic recrystallization during deformation of polycrystalline ice: Insights from numerical simulations. Philosophical Transactions of the Royal Society A: Mathematical, Physical and Engineering Sciences, 375 (2086), art. no. 20150346.

Llorens, M.-G., Griera, A., Bons, P.D., Roessiger, J., Lebensohn, R., Evans, L., Weikusat, I. (2016a) Dynamic recrystallisation of ice aggregates during co-axial viscoplastic deformation: A numerical approach. Journal of Glaciology, 62 (232), pp. 359-377.

Llorens, M.-G., Griera, A., Bons, P.D., Lebensohn, R.A., Evans, L.A., Jansen, D., Weikusat, I. (2016b) Full-field predictions of ice dynamic recrystallisation under simple shear conditions. Earth and Planetary Science Letters, 450, pp. 233-242.

Steinbach, F., Kuiper, E.-J.N., Eichler, J., Bons, P.D., Drury, M.R., Griera, A., Pennock, G.M., Weikusat, I.(2017) The relevance of grain dissection for grain size reduction in polar ice: insights from numerical models and ice core microstructure analysis. Frontiers in Earth Science, 5, art. no. 66.
* * *

---

## Author Comment (AC1) · 27 Feb 2019

We thanks the reviewer for the in depth comments we received. Here is a detailed response for all addressed comments, along with an attached revised version of the Manuscript and Supplementary.

Specific comment 1: A deeper discussion of grain-size sensitive deformation mechanisms may be appropriate. For example, the different nature of the distribution of GND's near grain boundaries is an interesting and important observation. One interpretation of this observation is that there is a heterogenous distribution in the magnitude and orientation of stress near grain boundaries. In other words, the presence of grain boundaries may enhance deformation, which would lead to a grain-size sensitive rheological behaviour such as described by Goldsby and Kohlstedt (2001). A grain-size sensitive rheological behaviour may operate at the higher strain conditions where weakening (increasing strain rate with time) is observed. Although it is possible that the weakening is entirely due to the alignment of grains with favourable orientation to operate easy slip at larger shear strain (i.e. geometric softening).

Response : In the conditions prevailing in the experiments presented in this article, that are, plurimilimetric grains, rather low stresses, and temperatures very close to Tm, the grain-size sensitive regime is not concerned. This assumption is corroborated by previous experiments under axial compression using similar microstructures and pressure and temperature conditions, which do not show any grain size dependence (Schulson and Duval, 2009, Montagnat et al., 2015) Indeed, the GSS regime prevails in areas of very small, micron-sized grains, and under conditions where strain accommodation mechanisms such as dynamic recrystallization are inefficient (because diffusion at the characteristic length scales of the system is too slow, see for instance the recent work of Bourcier et al. 2013 on halite). Grain boundary sliding has been observed in ice with large grains at faster strain-rates of about 10-5 s-1 (Weiss and Schulson 2000), but it occurred just prior to cracking and was accompanied by a large amount of decohesion that we do not observe in the present experiments. The analysis of the microstructure clearly indicates that in the torsion experiments presented here, dynamic recrystallization mechanisms (especially grain boundary migration) are very efficient, since we are close to Tm. Efficient dynamic recrystallization relaxes the local stress and strain heterogeneities, which are caused by the viscoplastic anisotropy of the ice crystals and, as expected are stronger in the vicinity of boundaries between differently oriented grains, well before grain boundary sliding occurs. The development of strong heterogeneity in the strain and stress field controlling the onset of the dynamic recrystallization observed in the present experiments is a direct consequence of

deformation mostly produced by highly anisotropic dislocation motions within the crystals. The early onset of and widespread occurrence of dynamic recrystallization in our samples is an evidence of deformation mainly accommodated by dislocation motions. Indeed, the development of strong heterogeneity in the strain and stress field, which controls the dynamic recrystallization, is a direct consequence of deformation mostly produced by highly anisotropic dislocation motions within the crystals. Grain boundary sliding, if associated with effective accommodation of local strain incompatibility by diffusion creep or even by dislocation glide, should produce a much more homogeneous microstructure than the one we observed. In the present experiments, the strain distribution within the grains is heterogeneous, as a consequence of stress and strain heterogeneities, in particular when neighboring grains have contrasted behaviors due to different crystallographic orientations. On the other hand, there is no clear evidence for stress concentrations at triple junctions or at irregularities of the grain boundaries, which are expected in the case of grain boundary sliding. In summary, although grain boundary migration is certainly important as a strain accommodation process, all observations points to deformation mainly accommodated by intragranular dislocation motions, which is grain-size independent. Grain boundary migration is certainly affected by the gradient in dislocation density characterized in the present work. This gradient result probably in a characteristic length scale for formation of nucleii, as discussed in the article. However, this length scale is independent of the size of the deformed grains. Thus, we consider that a lengthy discussion of the grain-size sensitive regime is out of the scope of this article. Concerning the mechanical data, as pointed by the reviewer and shown in a large number of previous experiments studying tertiary creep (e.g., Bouchez and Duval, 1982; Jacka and Jun 1994), the texture-related weakening may easily explain the onset of tertiary creep, during which a constant grain size is observed owing to the balance between nucleation and grain growth by boundary migration.

References: [1] E. M. Schulson and P. Duval. Creep and Fracture of Ice. Cambridge University Press, 2009.

[Figure]

[1] Montagnat, M., T. Chauve, F. Barou, A. Tommasi, B. Beausir, and C. Fressengeas. 2015. "Analysis of Dynamic Recrystallization of Ice from EBSD Orientation Mapping." Frontiers in Earth Sciences. http://www.gm.univ-montp2.fr/PERSO/tommasi/publications.html.

[2] Bourcier, M., M. Bornert, A. Dimanov, E. Héripré, and J. L. Raphanel. 2013. "Multiscale Experimental Investigation of Crystal Plasticity and Grain Boundary Sliding in Synthetic Halite Using Digital Image Correlation." Journal of Geophysical Research: Solid Earth 118 (2): 511–26. https://doi.org/10.1002/jgrb.50065.

[3] Weiss, J., and E. M. Schulson. 2000. "Grain-Boundary Sliding and Crack Nucleation in Ice." Philosophical Magazine A 80 (2): 279–300. https://doi.org/10.1080/01418610008212053

[4] Bouchez, J. l., and Paul Duval. 1982. "The Fabric of Polycrystalline Ice Deformed in Simple Shear: Experiments in Torsion, Natural Deformation and Geometrical Interpretation." Textures and Microstructures, 5: 171–90. https://doi.org/10.1155/TSM.5.171.

[5] T. H. Jacka and L. Jun. The steady-state crystal size of deforming ice. Ann. Glaciol., 20:13–18, 1994.

Specific comment 2: A stress exponent of 3 is used to estimate the stress at the outside radius of the samples. Although probably appropriate for this study, it would be better to have a stronger justification for this selection. For example, if a grain-size sensitive mechanism is operation at higher strain (smaller grain size) conditions, a lower value of stress exponent may be more appropriate. Additionally, values of n of 4 have been observed in ice deforming by dislocation creep (e.g., Durham et al., 1983). A stronger justification for using n=3 would place more confidence in your calculated values of stress and make future comparison of this work more straightforward.

Response : The value of 3 for the stress exponent was chosen considering that most of our experiments are close to the secondary creep conditions. From previous work

(a good review can be found in Shulson and Duval 2009), it was shown that during secondary creep (in fact at the minimum creep rate), the stress exponent for coarse-grained polycrystalline ice is close to 3. These studies also show that during tertiary creep this exponent increase, reaching a value close to 4. These early data is corroborated by recent work of Treverrow et al. (2012) presenting results from compression and shear tests, which observed stress exponents between 2.9 and 3.1 for the secondary creep, and of 3.5 for tertiary creep. Reference to this study has been added in the new version of the paper to justify the choice of the value of 3.

Modified text (P.4 L.28) "Since most of the present experiments recorded secondary creep conditions (Fig. 1), stress exponent of n = 3 was chosen based on results from Duval et al. (1983) and more recent work from Treverrow et al. (2012) that observed stress exponents between 2.9 and 3.1 during the secondary creep regime, and of 3.5 for tertiary creep in compression and shear tests.

References: [1] Schulson, Erland M., and Paul Duval. 2009. Creep and Fracture of Ice. 1 edition. Cambridge, UK ; New York: Cambridge University Press.

[2] Treverrow, A., et al., Journal of Glaciology 58, no. 208 (ed 2012): 301–14. https://doi.org/10.3189/2012JoG11J149.

Specific comment 3: A discussion of some more of the recent work on the controls of ice CPO at low strain conditions would help strengthen some of the arguments in this manuscript. For example, Qi et al. (2017, JGR), discussed the importance of stress on controlling the nature of CPO in ice, and noted the importance of grain boundary migration at low stress and lattice rotation at high stress. Although those experiments were carried out at different conditions and using a different deformation geometry, they may provide some insight into the various conditions at which the texture with the M2 maxima are important.

Response : In the revised version of the manuscript, we include a comparison with both the work of Qi et al. (2017) and the more recent study of Qi et al. (2019). This comparison is interesting, since their experimental conditions are very different from the present ones: grain sizes are smaller, strain rates are faster, and stresses are higher. The confinement pressure (10 MPa in Qi et al. 2017) might also have some impact on the recrystallization mechanisms and texture evolution. Qi et al. 2017 presents results for axial compression. Studies in a similar highly anisotropic material, olivine, clearly indicate that the effect of recrystallization on the texture evolution is much more marked in simple shear (cf. review in Signorelli & Tommasi 2015). Thus we also compare our data to the very recent Qi et al. (2019) study, which presents data obtained in direct simple shear, which can be more easily compared to the present study, although some shortening normal to the shear plane (transpression) usually takes place in this configuration, as recognized by Qi et al. (2019). As stated in the revised version of the ms. the results of the three studies share many common points.

References : [1] Signorelli, Javier, and Andréa Tommasi. 2015. "Modeling the Effect of Subgrain Rotation Recrystallization on the Evolution of Olivine Crystal Preferred Orientations in Simple Shear." Earth and Planetary Science Letters 430 (November): 356–66. https://doi.org/10.1016/j.epsl.2015.08.018.

Specific comment 4: Sample TGI0.012 was deformed to very low strain and its data are missing from Figure 1. How certain are you that this small amount of plastic strain was imposed on the sample? The grain size of TSGI0.012 appears smaller than the unstrained sample, which suggests at least some plastic deformation occurred. Was a correction made for the compliance of the rig? Was there any evidence of elastic strain?

Response : About the elastic compliance of the rig, previous work on ice single crystals had shown that the strain measurement of the torsion apparatus used here is accurate even at very low strain (Montagnat et al. 2006, Chevy et al. 2010). Although small, this sample (TGI0.012) was affected by viscoplastic strain, not only by elastic deformation. Indeed, given the value of the shear modulus for ice, the elastic deformation is really small. At the maximum shear stress of 0.6MPa, the shear modulus of 3.5 GPa leads

to $\gamma$elastic of about 2.10-4. In addition, although subtle, the change in the texture relatively to the undeformed sample suggests that it experimented a small amount of viscoplastic strain.

As for the variation in grain size, different samples were used for each test, and owing to the sample growth procedure, a slight difference in initial grain size can occur. The difference between the grain size of the "undeformed" microstructure shown in the paper for illustration, and the one of the TGI0.012 is in the range of usual variation due to the preparation procedure. In any case, the very small strain that affected this sample did not trigger significant changes in the microstructure and in the texture. By consequence, this sample does not play a major role on the conclusions of the present study. It gives nevertheless a control on the very first stages of the deformation.

References: [1] M. Montagnat, J. Weiss, J. Chevy, P. Duval, H. Brunjail, P. Bastie, and J. Gil Sevillano. The heteroge- neous nature of slip in ice single crystals deformed under torsion. Philosophical Magazine, 86(27):4259– 4270, 2006. [2] J. Chevy, C. Fressengeas, M. Lebyodkin, V. Taupin, P. Bastie, and P. Duval. Characterizing short-range vs. long-range spatial correlations in dislocation distributions. Acta Materialia, 58(5):1837 – 1849, 2010.

Technical Corrections We thank the Reviewer for the detailed correction of the manuscript. The updated version of the manuscript has been corrected accordingly, a response to a specific comment is given hereafter otherwise.

p2 ligne 3 : "compression and extension are the dominant deformation mecha- nisms" – this is a bit awkward as most people discussion deformation mechanisms as related to flow behaviour, e.g., dislocation creep, diffusion creep, etc. It may be more clear to replace "deformation mechanisms" with "deformation geometries". R : done

Line 6: remove "s" from "orientations" R : done

Line 10: remove "olivine" R : done

Line 23: replace "in return" with "consequentially" R : done

Line 27: add "rate" after "strain" ? R : done

Lines 28 – Page 3, line 2 – The statements made in this paragraph are a bit debatable. I am not sure if ice is a great analogue for mantle rocks. Hundreds (if not thousands) of high temperature experiments have been carried out to study the flow behaviour and microstructural characteristics of mantle rocks. Ice is significantly more anisotropic, in a viscous sense, than olivine and other mantle material.

Response : CPOs obtained under simple shear for ice and some minerals such as olivine or quartz are similar: in all cases the main slip system rotates into parallelism with the macroscopic strain much faster than the reorientation of the finite strain ellipsoid and the CPO reaches a stable position and intensity at rather low shear strains (2-5). By consequence, similar explanations (related to recrystallization) were proposed (see for instance Wenk et al. 1997, 1999 and Signoreli and Tommasi 2015). In terms of viscoplastic anisotropy, both ice and olivine do not possess enough available slip systems for accommodating a general deformation - in particular shortening or extension parallel to the [100], [010], and [001] axes in olivine and parallel to [0001] in ice (see for instance Castelnau et al. 2008). Comparing the intensity of anisotropy of the two materials is not straightforward. Ice has a higher symmetry (hexagonal): slip is only easy in the basal plane, but it shows with no anisotropy within this plane. Olivine is orthorhombic: it has more slip planes, but only two slip directions in these planes. In conclusion, both are extremely anisotropic materials. This leads to strain compatibility problems and stress concentrations in the polycrystal, which, at high homologous temperature, are resolved by recrystallization.

References: [1] H. R. Wenk, G. Canova, Y. Bréchet, and L. Flandin. A deformation-based model for recrystallization of anisotropic materials. Acta. Mater., 45(8):3283–3296, 1997. [1] H. R. Wenk and C. Tomé. Modeling dynamic recrystallization of olivine aggregates deformed in simple shear. J. Geophys. Res., 104(B11):25,513–

25,527, 1999. [3] Signorelli, J. and Tommasi, A.: Modeling the effect of sub-grain rotation recrystallization on the evolution of olivine crystal preferred orien-tations in simple shear, Earth and Planetary Science Letters, 430, 356–366, https://doi.org/10.1016/j.epsl.2015.08.018, 2015 [4] O. Castelnau, D. K. Blackman, R. A. Lebensohn, and P. Ponte-Castaneda. Micromechanical modeling of the viscoplastic behavior of olivine. Journal of Geophysical Research, 113:B09202, 2008.

Page 3: line 10: remove "has" R : done Line 28: add "either" before "not" R : done Line 29: remove space after "mechanisms" R : done Page 4 Line 6: replace "packing evenly" with "evenly packing" R : done Line 17: replace "control visually" with "allow for observation of" R : done Page 5 Line 18: replace "does" with "do" R : done Line 31: replace "identify" with "identified" R : done Page 6 Line 20: replace "reminded" with "noted" R : done

Page 7 Line 1: here you say the noise was too large to distinguish any primary creep hardening in TGI0.012 R : See response to Line 5-6

Line1: add space after at end of sentence R : done Line 5-6: Can you say for certain the TGI0.012 stayed in the primary creep regime? The strain was very small and the data for that experiment is not presented in Figure 1 Response : This is an assumption, the text in the updated version of the manuscript has been modified to reflect the reviewer comment.

Line 21: add "essentially" before "random" R : done Line26-27: Did TGI0.012 really achieve the strain indicated? How did you account for compliance and elastic defor-mation of the sample? Response : Considering the value of the shear modulus for ice, elastic deformation is really small and cannot be measured by our sensor. Taking the max shear stress of 0.6MPa, the shear modulus of 3.5 GPa leads to gamma_el of about 2.10-4.

Page 8 Figure 1: missing data for TGI0.012 R : Data for TGI0.012 is too short in timescale to be practical to show in figure 1.

Figure 1 caption: replace "experiments" with "experiments", or change this sentence unless you add data for TGI0.012 to the figure. Replace "The blank part of the curve corresponds to" with "the blank parts of some of the curves correspond to". Remove "represented". R : done

Page 9 Line 10: add "(by gamma = 0.2) R : done

Page 10 Figure 3: This is a nice plot but the large gap between gamma 1 and 2 suggests that maybe it is worthwhile mapping an axial section of TGI1.96? This would allow you to calculate the J-index at all strains by mapping form the center (almost unstrained) to the outside (almost gamma=2) of the sample. R : One axial section was measured with EBSD, but due to the coarse grain size, it was impossible to fractionate the section in different segments with 'supposed' constant finite shear strains and still have a high enough number of measurements to obtain representative estimates of the CPO intensity. We could not therefore use radial sections to estimate the CPO evolution with increasing strain.

Line 1: replace "develops" with "develop" R : done Line 9: remove sentence about spatial resolution, that is already in the methods section R : done Page 12 Figure 5: what is represented by the pole figures? c-axes? R : Text has been added to the caption : Pole figures representing c-axis orientations are reported at the top right. Line 2: add "dislocations" after "those" R : done Line 9: add "investigated" after "strain" R : done Page 13 Line 3: replace "identify" with "identified" R : done Line 5: replace "kinds" with "types" R : done Line 16: remove " t "R : done Page 17 Line 15: replace "these study" with "those studies" R : done Page 19 Line 7: Replace "advices" with "advice" R : done

Please also note the supplement to this comment:
https://www.the-cryosphere-discuss.net/tc-2018-213/tc-2018-213-AC1-supplement.zip

---

## Author Comment (AC2) · 27 Feb 2019

We thank the reviewer for the detailed comments as we think it helped to greatly enhance the quality of our manuscript. We provide here answers to all the reviewer's comments, along with the revised manuscript and Supplementary materials.

Scientific Discussion 1.a: The <11-20> and <10-10> in the high strain sample ($\gamma$=1.96) are not randomly distributed within the girdle. The <11-20> and <10-10> both have broad maxima, parallel to the shear direction, of $\sim$ 4x m.u.d. and $\sim$3 x m.u.d. respectively. These compare to minima within the girdle of ∼ 2x m.u.d. This level of <a> and <m> alignment is comparable to that shown for the highest shear strain data at -5C in fig 4 of (Qi et al., 2018). Additionally the ratio to the <c> axis maximum (max <a> ∼ max (c)/y where y is between 2 and 4) is very similar to the highest shear strain data at -5C and all data at -20C and -30C in fig 4 of (Qi et al., 2018). The alignment of <a> and <m> orientations is important. This might provide a cool tool for assessing shear directions in the analysis of naturally deformed ice so it needs to be documented. <a> and <m> being co-aligned matches our data and is intriguing. At present I do not have a coherent explanation for this. I'd be interested to hear your views on this.

Response : Our data presents indeed a preferred orientation of <a> and <m> directions within the girdle, as reported by Qi et al. (2018), instead of being randomly distributed within the girdle. We edited the figure 4 to adapt the ODF intensity to make the distribution easier to see, and added the following description in the manuscript:

Added Text (pp.11 L.11): For the high shear strain sample at $\gamma$max=1.96, the <10-10> and <11-20> axes (<a> and <m> axis) form a girdle, which tends to align in the shear plane. Within this girdle, there is a preferred orientation of both <a> and <m> directions toward the shear direction. The present CPO is similar to those formed in direct shear experiments (Qi et al., 2019). It is consistent with equivalent contribution of the three <a> axes in accommodating shear on the (0001) plane.

Scientific Discussion 1.b: You have not commented on the shape of the M1 and M2 maxima. In virtually all experimentally sheared polycrystalline ice samples these maxima are elongated in a direction perpendicular to the shear direction (see discussion in (Qi et al., 2018) and in our response to a Maurine Montagnat comment on this in the discussion section). Sometimes the elongated maxima (both M1 and M2) are actually each double maxima, with the profile plane as a mirror plane. The vast majority of naturally sheared ice samples do not have elongated maxima, the contours of the maxima match small circle distributions (e.g. (Hudleston, 1977)). This point of difference between experiment and nature is important and as such it is important that the shape

of the M1 and M2 maxima from experiments is described. The high strain ($\gamma$=1.96) M1 is clearly elongated in the direction perpendicular to shear. I have superposed small circles, with their cone axes on the primitive, on the figure above to emphasise this point. M1 in the lower strain experiments is not so clearly elongated. In the annealed experiment the contours match the small circles, and it looks like this is the case for the lower strain experiments. In our experiments (Qi et al., 2018) elongation increases with shear strain. M2 in the $\gamma$=0.42 experiment is elongated, with a double maximum (labeled above max1, max2), with the profile plane as a mirror plane. The $\gamma$=0.42 experiment may also show this but I can't tell from the figure. Interestingly M2 in the annealed sample does not look elongated. This could be an important point. Does annealing remove the cluster elongation? One of the reasons we adopted a different reference frame in (Qi et al., 2018), with the pole to shear plane in the middle of the stereonet, is that it makes it easier to see cluster shapes, as shown below in a re-analysis of the (Bouchez and Duval, 1982) data. The highest and lowest strain samples in these data have elongated M1, the medium strain sample does not.

Response : The reviewer pointed out an aspect of our results that was understated in the initial version of the ms. We included the following text in the manuscript:

Added Text (pp.11 L.16): Some elongation of the distribution of the M1 and M2 sub-maxima towards the Z direction (Figure 4.b), which is the normal to the shear direction in the shear plane, is visible in our results. This elongation is best expressed for the M1 maximum in the highest strain sample TG1.96, for which pole figures for <0001> <10-10> and <11-20> lattice vectors are now represented in two perpendicular reference frame in figure 4.b for better readability. Similar elongated distributions of <c> axes have been reported in direct shear experiments by Qi et al. (2019). Some elongation of the M1 maximum is also observed in the highest shear strain sample (gamma = 2) of Bouchez and Duval (1982) as well as in other shear experiments in Li et al. (2000), Wilson and Peternell (2012) and Budd et al. (2013). However, most naturally sheared ice samples do not have elongated <c> maxima (Hudleston, 1977).

Scientific Discussion 1.c: I think you need to be a little more precise in description of the symmetry of the M1, M2 maxima pair with respect to the finite elongation direction. I think this is a cool observation and potentially of some value, but the symmetry is far from perfect. Below I have plotted up some traces for M1 and M2 (red lines), with angles measured from the top of the stereonet. The green line has equal angles to the two red traces. Superficially this green line is close to the finite extension direction (ED), but if I plot the expected M2 trace (yellow line) assuming it has the same angle to ED as M1 (and adjusting ED for for M1 not being at 0 degrees in the two lowest strains) then the observed M2 is anticlockwise of the yellow line for the three lowest strain, most markedly for the annealed sample. The symmetry you describe is approximate. Another way of looking at this is to plot the angle between M1 and M2 against shear strain. Below is a modified version of fig 8 from (Qi et al., 2018) with the addition of your data (big red dots) and a line (pink) that predicts the position of M2 if it has the same angle to the finite extension direction as M1. This is quite an interesting addition to the plot as very broadly the red data points (high T experiments: not just yours) do follow the path of the pink line, but at slightly lower angles? Is M2 at high T and low shear strain ($<=\sim2$) related to the orientation of the finite strain ellipsoid?

Response: The reviewer was right in noticing that the symmetry is not perfect. We included in figure 2.b the directions and angles of the M1 and M2 submaxima, along with the phi angle, for ease comparison with results from Qi et al. (2019). We also reported in the manuscript that the symmetry between M1 and M2 relative to the finite extension direction was not perfect. M2 maxima are at larger angles to the finite extension direction (ED) than what would be expected for a perfect symmetry. Nevertheless, the residual M2 angle relatively to the perfect symmetrical direction with M1 is rather small (between 1 and $3°$), except for the annealed sample TGI0.71, which has an angle of $17°$. This deviation could be due to annealing processes and is now discussed in a bit more details in the discussion of the new manuscript. Since the number of experiments performed in the present study is too small for a statistical analysis, we prefer not to discuss this point into more detail in the current manuscript. We refer therefore the

interested reader to the discussion in Qi et al. (2019) paper for more details on this point. We would be happy in any case to share our data, if the reviewer was willing to use it to complete his data set, as exemplified in the figure presented in the review.

Added Text (pp.10 L.1): "Nevertheless the symmetry between M1 and M2 around the finite extension direction is not perfect. The angle between M2 and ED is generally larger than the angle between M1 and ED by 1 to 3âŮę. The exception is the annealed sample TGI0.71, where the difference between the two is 17.4âŮę, with M2 closer to ED than it should have been for a perfect symmetry. This change may be due to a post-deformation CPO evolution, due to grain growth during annealing. A small lag in the reorientation of the M2 submaximum relative to the M1 submaximum is also observed in other simple shear experiments (Bouchez and Duval, 1982; Qi et al., 2019). The limited number of experiments performed in the present study precludes a statistical analysis of this behavior. The evolution of the angle between M1 and M2 ($\varphi$) with increasing shear strain is, nevertheless, discussed in more detail in Qi et al. (2019), which present a comparison between observations in ice shear experiments results at different temperatures and numerical modeling.

Added Text (pp.17 L.17): "Furthermore we report an offset angle between the M1 and M2 submaxima of 17.4âŮę greater than what would be expected for a perfect symmetry to the finite extension direction. Other experiments, which didn't undergo annealing show a difference in angle of only 1-3âŮę with the perfect symetry. In the TGI0.71 annealed sample, M2 is much closer to M1, as would be expected for a higher finite shear strain. This could be interpreted as a sign of preferential growth of bulging nucleii with orientation closer to M1 than the bulk CPO of the sample before annealing. "

Scientific Discussion 2 The description/ documentation of the experimental set up needs to be improved. Please provide some key diagrams that show the experimental set up. Torsion is an important deformation kinematic and the torsion experiments you show here and the classic work of (Bouchez and Duval, 1982) represent significant

contributions to our understanding of ice with direct application to polar ice sheets and glaciers. I believe that torsion is an important defomation kinematic to explore more fully in the future. The picture in (Duval, 1976) and the words in (Bouchez and Duval, 1982), (Duval, 1976) and presented here are insufficient for someone to reproduce the experimental set up. It would be great if you could present (maybe in supplementary information) some diagrams that show the mechanics of the deformation apparatus. There is one particular aspect that I think is of paramount importance. I think that this apparatus is constrained to deliver simple shear, with no shortening or extension normal to the shear plane. If this is the case I presume that the "platens", that deliver the torque, are fixed so that they cannot move normal to the shear plane. This is important so that we can be clear which experiments are simple shear only, and which comprise simple shear with a component of shortening (or extension). This is not necessarily the same as having zero normal stress on the shear plane. (Li et al., 2000) (a key paper that is not cited in your work) point out that direct shear experiments using a "Jacka" rig, with the normal load set as zero still experience shortening/ extension normal to the shear plane (and that the magnitude depends on sample geometry). Furthermore they suggest that an experiment with fixed platens will generate shear plane normal stresses of 0.1 to 0.2MPa. In my view a constrained (by fixed platens) simple shear experiment is great - it's a clear kinematic end member. We do need to be absolutely clear about the experimental kinematics and the implications the kinematics have for stress, rheology and microstructure. What are the kinematics and dynamics of naturally deforming ice systems is yet another matter. I can imagine some scenarios (e.g. ice stream margins) where perfect simple shear may occur and others (e.g. basal zones) where shear with shortening parallel to the shear plane occurs.

Response: We added a scheme of the experimental setup in the supplementary material. The presented experiment is indeed a simple shear setup with fixed plates. The reviewer is right, the "platens", that deliver the torque, are fixed so that they cannot move normal to the shear plane. Furthermore, the sample is held horizontally. Therefore, no extension of compression component of stress are delivered.

Added text : (pp.4 L.18): The design of the torsion apparatus does not allow for displacements parallel to the rotation axis; the imposed deformation is therefore perfect simple shear. During the experiments, the evolution of the CPO under these fixed-end boundary conditions might produce axial stresses (Swift, 1947). The latter cannot be measured in the present apparatus, but polycrystal plasticity models indicate that these axial stresses may attain values similar to those of the shear stresses when the CPO is oblique to the imposed shear (Castelnau et al., 1996). A more precise description of the apparatus is provided in Supplementary.

Scientific Discussion 3 The mechanical data are a bit puzzling. The focus of this paper is the microstructure, and I don't think the questions about the mechanical data affects substantially the microstructural observations and interpretations, but I would like to see a bit more analysis of the data. The key problem for me is that the applied shear stress should be the dominant control of the shear strain rate (whether secondary, tertiary of at a $\sim$ given strain in transient creep), given that your temperature and starting materials were nominally the same for all experiments. A shear stress of 0.6MPa vs 0.50.5MPa should give a $\sim$ doubling of strain rate (for n between 3 and 4). The secondary creep rate for TG10.42 (0.6MPa) is slower than that for TG10.71 (0.5MPa) and faster than TG10.2 (also 0.5MPa). In the text this is attributed to "variability of grain size and textures". This could be true, but in needs to be unpicked in a bit more detail. The method used to fabricate the starting material sounds the same as that we use (except that we do not anneal) as described in (Stern et al., 1997). We have looked at >10 samples of starting materials made by the same methods in four different labs (Otago, MIT, UPenn, UCL) and all have very very similar grain size distributions, mean grain size and random CPO; an example is in fig 1a in (Qi et al., 2017). I cannot see that the annealing will affect the CPO and annealing at consistent T and time should give the same grain size distribution. Do you have initial g size data from more than one sample? We can estimate what grain size differences would be needed to explain the variations in secondary creep rate. The ratio of secondary creep rates of the two samples deformed at 0.5MPa is about 2 (estimated from slopes on fig: would be good

to provide an enlargement of secondary creep region, as you have done for primary creep region). Using the grain size exponent (-1.4) from (Goldsby, 2006; Goldsby and Kohlstedt, 2001)jb this would require the relative mean grain sizes of the two samples to be $\sim$ 1.7. (e.g 1.5mm and 0.9mm). This grain size exponent may be a bit large. A more conservative estimate (related to similar starting materials) comes from using the peak stress (= secondary creep) data in (Qi et al., 2017), fig 3. This gives an $\sim$ grain size exponent of -0.8, requiring a grain size ratio of $\sim$2.3 (e.g 1.5mm and 0.65mm) to explain the strain rate differences at 0.5MPa. I am pretty sure that your original grain sizes do not vary by a factor of $\sim$2, so grain size is unlikely to provide an explanation for the variability in mechanical data. Although it seems likely that your bulk CPO is random in all starting materials, it is worth considering whether the sample cross section contains enough grains to give the mechanical properties of a random CPO. This was clearly an issue for us deforming 1 inch diameter samples with a $\sim$5mm grain size (Craw et al., 2018): in this case a cross section may contain only 10 or 20 grains and the peak stress (= secondary minimum) data do not have a systematic relationship to strain rate. In your case there should be $\sim$ 500 grains in a 35mm diameter cross-section so I would have thought this effect is unlikely to be significant. It seems unlikely to me that the variations in strain rate relate to variability in the starting material. In this case it's worth looking back at the experimental set up. How is stress transferred from the rotational drive platens (this needs describing- see point 2) to the sample? Is there a possibility that there is some slippage (frictional loss) or other parameter that varies from one sample to the next so that the torque is not all transferred to shear stress on the sample?

Response: We have been a bit fast in proposing that initial grain size and texture could be at the origin of the difference in mechanical response between our different tests. In fact, although some slight variations of those two parameters are expected to occur from sample to sample, they are probably too small to justify the measured differences in strain rate. Unfortunately the starting CPO and grain size distribution was not measured for each sample. Nevertheless, we thank the reviewer for the comment

on the number of grain limitation. Indeed, with a starting grain size of 0.7 mm we can expect 45-50 grains in diameter in our samples (and not 500). If we talk in terms of radii (were the gradient in shear is applied, the picture gets even worse, with only 23-25 grains. This could indeed have as strong influence on the strain rate which could explain the difference we see here. We included in the new version of the manuscript a note to use these curves with cautions because of the points discussed above.

Added text (pp.7 L.19): The significant variations observed in strain rate evolution with time between the different runs cannot be attributed to a variation in initial grain size, CPOs or in the applied torque alone (Table 1), but rather to coarse-grained microstructure of the samples, which resulted in less than 25 per radii. The strain/time curves presented in figure 1 are therefore useful to characterize each run creep regime independently, bur should be used with care in comparison between different samples or with other experiments.

Scientific Discussion 4 The discussion of modeling is rather black and white and superficial. Numerical models and physical experiments all have limiting boundary conditions. All models and experiments show us something and none match nature, primarily because we cannot access natural conditions and have uncertainties about natural boundary conditions. Linking physical experiments to numerical models is important as we have much more control on the boundary conditions in both cases: so we learn more about our understanding of processes. However the crucial thing for both experiments and models is that we are clear about what we learn from them. I think having a model that is able to simulate fully CPO and microstructure evolution at high strains is still a way off. All steps on the way to achieving this are valuable and a discussion that implicates that one model is right and another wrong is inappropriate: in demeans what we learn from the models. I agree that the Etchecopar model as used in (Bouchez and Duval, 1982) matches quite well your data and most of the "hot" shear data (see the red symbols in M1-M2 angle vs shear strain graph posted earlier: Etchecopar model also plotted on this as hollow black squares). The problem is that these are the only

data it fits, so if this model is applicable it tells us only part of the story. The model does not predict the drop off to single maxima by shear strain of 2 in (Li et al., 2000); maybe this is a kinematic difference between simple shear and simple shear plus some strain normal to shear. The model does not match the minimal "colder" data we have, most particularly the -30 data from (Qi et al., 2018). The FFT model (Lebensohn, 2001) gives a remarkable match to experimental observations of intragranular deformation at low strain (Grennerat et al., 2012; Lebensohn et al., 2009). This is the code used to simulate shear deformation in the models by (Llorens et al., 2016; Llorens et al., 2017). The fact that the same model works well at low strain and less well at high strain tells us something. The bulk CPOs in Lloren's models do not have double maxima, but the double maxima are there when only the high strain rate data are used (see Llorens, 2017 fig 5i) and the angle between maxima in the deformation only models evolves in a way that matches the -30 experimental data we have (Qi et al., 2018). Addition of recrystallization into the model changes the result, although not in a way that gives a really clear match to observations. There is no real conclusion here apart from this: both models and experiments are important. Probably most important is to design experiments that enable clear boundary condition matches to numerical models. That is the really beautiful thing about the columnar ice work at low strain e.g. (Grennerat et al., 2012). At high strain and in shear matching of model and experimental boundary conditions is rather harder.

Response: This response also takes into account the remark by Griera, Bons & Llorens,. We made use of the Etchecopar model just to highlight the likely role of subgrain boundaries in the process of accommodating basal glide of dislocations during simple shear of ice. To our point of view, this model can only explain that strain incompatibility accommodation processes are required to obtain the strong single maximum observed in the laboratory and in the field (Hudleston et al. 1977). We will be clearer about that in the text. Considering FFT homogenization schemes as the one used by Llorens et al. (2017) and the one that we used in Grennerat et al. (2012) or self-consistent viscoplastic models (Castelnau et al. 1996), they stand on strain

being produced by the activity of slip systems only. For ice, the only slip system for which there is experimental evidence of easy activation is the basal system. However, the basal slip system cannot, alone, produce a general type of deformation. Thus for maintaining strain compatibility, these homogenization approaches require the activation of the non-basal systems, namely, prismatic and pyramidal systems. Activation of these systems induces specific rotations of the crystals. This is the main reason why, unless extra mechanisms (which mimic the role of dynamic recrystallization in helping to enforce strain compatibility) are added to these models, such as in Wenk et al. (1999) or Signorelli and Tommasi (2015), the crystals never reach the stable position observed experimentally or in naturally deformed samples, in which the main slip system is parallel with the imposed macroscopic.. Models, which do not include any strain compatibility relaxation process, as Llorens et al. (2017) produce a strong clustering of c-axes, similar in intensity to the one observed experimentally or in the field, but offset from the normal to the shear plane. We do not pretend that polycrystal plasticity models are not useful. We just discuss that, by construction, the vertical single maximum cannot be reproduced in a model where deformation is fully accommodated by dislocation glide. In the experiments, other mechanisms do come into play. By consequence, yes, the comparison is very helpful to quantify the role of these mechanisms. The text has been modified to be more clear about this point. We also noticed a mistake. The reference about simple shear modeling of ice in simple shear is Castelnau et al; 1996, JGR. It has been modified in the text.

Added text: (pp.18 L15): Pioneering work on 2D modeling of polycrystalline aggregates under simple shear by Etchecopar (1977) was able to repro- duce the sub-maxima M1 and M2. This was simply done by considering a single slip system (basal slip system for ice) and adding an accommodation process by allowing cells to subdivide (polygonization) and undergo rigid body rotation. The very good agreement of this simplistic model with evolution of textures observed experimentally for ice under shear was empha- sized by Bouchez and Duval (1982), who hypothesized that the polygonization processes in ice would be formation of GNDs and kink-bands. In our results few kink

Interactive
comment

bands were observed, but the prevalence of GNDs at most finite shear strains suggests that Bouchez and Duval (1982) supposition is reasonable. Although Etchecopar (1977) is too simplistic to pretend reproduc- ing every shear-induced textures in ice, it was useful to raise the likely role of polygonization as an efficient accommodation mechanism for solving strain incompatibility problems. Modeling of shear in ice has been done by mean-field approaches as in (Castelnau et al., 1996) or more recently by full-field modeling as in (Llorens et al., 2016). Both works reproduced the formation of a strong single maximum texture from shear strain of about 0.4 and above. Nevertheless, neither orientation of this single maximum normal to the shear plane, nor the existence of two submaxima observed at lower strains in the field or experimentally are correctly reproduced. The fact that the single submaxima prescribed is inclined from the tangent to the shear plane is significant, and stands from the fact that these homogenization techniques require the activation of non-basal slip systems. The activation of secondary slip systems, whose contribution to strain has never been proven experimentally, induces a geometrical rotation of the crystal, that is responsible for the modeled inclination of the clustered CPO compared to the vertical. The activity of these secondary slip systems relative to the basal ones is controlled by a parameter that is arbitrarily defined (it has been defined in comparison to experimental observations in Castelnau et al. (1997), using the mean-field VPSC approach, and values different than the one chosen is the previously cited studies were obtained). The higher is non-basal activity, the softer is the mechanical response of the crystal to accommodate the imposed conditions. The geometrical constraint of crystal rotation under shear, owing to the activity of non-basal slip systems, can be artificially relaxed, such as in Wenk and TomeÌĄ (1999), by forcing the growth of selected grains, or as in Signorelli and Tommasi (2015), by an association of polygonization and local (within a grain) re-laxation of the strain compatibility constraints. By comparing these various modeling approaches, and their inclusion of recrystallization mechanisms, it appears that accommodation mechanisms, other than non-basal slip systems, must come into play to explain recrystallization induced shear textures in ice. Although we consider that fast

grain boundary migration might be an efficient strain accommodation mechanisms, we suggest here that an efficient additional contribution to the texture reorientation, at the high homologous temperatures of our experimental studies (and the ones of Bouchez and Duval (1982) or Qi et al. (2017, 2019)), might well be nucleation assisted by polygonisation (or sub-grain boundary rotation).

Clarity of writing 1/The bulk of the text is well-written. The clarity of the writing is not as good in the discussion and not good at all in the conclusions. The discussion would benefit from some shortening and restructuring. The discussion starts with a reminder of the key observational data and I think it would be very helpful to the reader if you added a schematic diagram to highlight these key observations. This would then give a clear framework for ongoing discussion. The conclusions needs to have clear statements on what are the new factual observations and what are the interpretations of those observations.

Response : We edited the discussion and conclusion to enhance the clarity. We have included bullet points in the conclusion

2/The abstract should be a concise summary of the new findings and some short statement about importance. The abstract contains an extended statement of background that is better placed in the introduction (it is in fact already in the introduction).

Response : We feel that a very short background in the abstract can be relevant for some readers not coming from an Earth Science or glaciology background. To shorten it we edited the abstract to remove the background statement on modeling.

3/I would go for a simpler title: "Evolution in polycrystalline ice microstructure during progressive high temperature shear" ????

Response : We thanks the reviewer for this suggestion and we changed the title to : "Recrystallization processes, microstructure and texture evolution in polycrystalline ice during high temperature simple shear"

[Figure]

Technical/ terminological/ picky things 5. It would be great if you could show full grain size distributions (frequency plots). You are correct that the mean is not a great scalar to represent recrystallized grain size statistics. Grain size distributions could be represented as an extra row in figs 2 and 4 (it would be nice to compare the AITA and EBSD measures- I don't expect them to be the same: see (Cross et al., 2017))

Response: During the analysis of the data we found that due to the small number of experimental runs (3 with gamma_max > 0.2 without annealing) made in this study the comparison of grain size frequency plots was not providing enough clear information to be included in the main manuscript. We have added the grain size frequency plots for both AITA and EBSD as supplementary material.

6. Please put the number of grains that correspond to each pole figure on figs 2 and 4 or in a table. This is important in comparing data sets.

Response: We have included the number of segmented grains in figure 2 and 4.

7. If you can, show point stereonets as well as contoured nets. The contoureing hides a lot of information.

Response: We added point stereonets in supplementary materials and kept the contoured ones in the main manuscript to maintain the readability of figures 2 and 4.

8. The statement on page 2, line 26 states that the "texture can increase shear strain rate (word "rate" missing) by a factor of ... ". There is a clear correlative relationship of weakening and CPO but a causative relationship is not established. Weakening in ice from secondary to tertiary creep correlates with development of a CPO. It is intuitive that the CPO developed in shear facilitates further shear. However similar weakening occurs in cold axial shortening where the CPO (cluster of c-axes parallel to shortening) would intuitively make further axial shortening harder e.g. -30 experiments right hand column of fig 3 in (Craw et al., 2018), mechanical data in fig 10. Other changes correlate with weakening, most particularly grain size changes (as documented in your

paper and elsewhere). In the geological literature grain size reduction is often thought of as the main cause of weakening. In reality CPO, grain size and other microstructural parameters all change in correlation to change in mechanical behavior. It is unlikely that the mechanical evolution is caused by changes to just one of these sample parameters.

Response: We agree with the reviewer and have changed the text accordingly to underline that weakening does correlate with the evolution CPO as well as other factors like the grain size.

9. I don't think that Kamb's idea that CPO is independent of of T, strain rate or stress is confirmed (P3, L11). The data in (Qi et al., 2018) show that in shear the CPO changes with T. (Qi et al., 2017) show that in axial shortening CPO is sensitive to stress or strain rate (the two cannot be separated). It is reasonable that the stress/ rate effect will also apply in shear. Using Huddleston's data in comparison to experiments is complex as both T and rate change. The lower rate has a similar effect to deforming hotter.

Response: Kamb (1972) states on his simple shear results on pp.233 : "Texture is sensitive to temperature, whereas fabric is not: recrystallization gives a distinctly coarser texture at the melting point than at temperatures only a few degrees below, whereas the fabrics developed under the two conditions are nearly the same." With the "texture" corresponding to the geoscience definition of grain shape and spatial relationships of grains, and "fabric" to CPO. We agree with the reviewer nonetheless that this conclusion is based on a small amount of results and on a limited temperature range (0 to -4°C in Kamb's work) and we rephrased this part of the text to address this point and refer to Qi et al. (2017,2019) work that shows a temperature and stress dependence of the CPO in both uniaxial and direct shear experiments.

Added text (pp.3 L7): Most of the knowledge on the microscopic processes occurring in polycrystalline ice under simple shear deformation is still mostly limited to deformation results from to data published over 30 years ago (Kamb, 1959, 1972; Duval, 1981;

Bouchez and Duval, 1982; Burg et al., 1986). The tools and methods used in these studies to analyze the CPO were often manual and highly dependent on the operator experience. Electron Back-Scattered Diffraction (EBSD) and Automatic Ice CPO Analyzer (AITA) can now provide high spatial and angular resolution quantitative data, enabling a global and statistical study of the processes accommodating strain at the micro-scale. Recent experiments (Qi et al., 2017, 2019) using these new characterization techniques have shed new light in some aspects of the question. They have, for instance, disproven the hypothesis by Kamb (1972) that CPO evolution in ice mainly depends on the finite shear strain and is not sensitive to temperature, strain rate, or stress. Indeed, Qi et al. (2017) that showed that during axial compression the final CPO is sensitive to stress or strain rate, and by Qi et al. (2019) which showed that the rate of evolution of the CPO in simple shear is sensitive to temperature.

10. The statement on page 5, line 8 is incorrect. Cryo EBSD of ice is not (in general) limited to samples of ∼10 by 20mm. In terms of published data there is a map in (Prior et al., 2015) (fig 12) of 80 by 30mm, the data in (Wongpan et al., 2018) has maps up to 40 by 40mm etc. Most of the CPO data we publish from experimental samples come from 25.4 by 40mm samples, our shear data CPOs in (Qi et al., 2018) are from elliptical shear surfaces of 25 by ∼ 30mm. For natural samples we routinely work on samples of ∼60 by 40mm and with suitable cold stage modifications I don't see why 100 by 50mm is not achievable. EBSD maps with the same dimensions as your AITA maps are possible now. If the Montpelier machine has a sample size limitation and this limitation is important to the paper, then link the limitation to that instrument, otherwise just delete the statement about size limitation. I guess if it the Montpelier machine does have a limitation it must be to do with cold stage tethering (gas pipes) or camera position limiting WD, as the sub-stage is designed for very large stages/samples (Seward et al., 2002).

Response: The reviewer is right and we changed the text accordingly.

11. Please provide enough information for the reader to understand how surface sublimation is managed. What I mean by this is; how is frost removed from the sample. There will be a frost layer on the sample surface as it goes into the SEM that would prevent EBSD (needs only ~ 10-20nm to do this). The two main ways of removing the frost are to heat the stage (Iliescu et al., 2004; Weikusat et al., 2011) or to cycle through pressure (Prior et al., 2015). I recall Andrea Tommasi telling me that the sample is just put in the SEM and it works. In this case I infer that the sublimation to remove the frost occurs on the down pressure cycle and that the sample is warm enough when put in the SEM to give a path through PT space where the sample goes into the vapour field (see fig7 in (Prior et al., 2015). In this case it would be useful to know the sample temperature on insertion and the pressure sequence: do you go to high vacuum then to controlled gas pressure or directly to controlled gas pressure?

Response: We use a different technique for surface preparation than the ones described by the reviewer, where we carefully remove the initial frost by carefully shaving the surface of the sample using microtome blades at -60°C before rapidly putting the sample in the SEM. We do not cycle in pressure or temperature and we never observed any issue with either sublimation or frost if we keep the sample below the sublimation temperature, which is at -60.6°C at 1 Pa. We feel that we provided already all the details in the manuscript and also in other manuscripts like Montagnat et al. (2015).

References: Montagnat, M., T. Chauve, F. Barou, A. Tommasi, B. Beausir, and C. Fressengeas. 2015. "Analysis of Dynamic Recrystallization of Ice from EBSD Orientation Mapping." Frontiers in Earth Sciences. http://www.gm.univ-montp2.fr/PERSO/tommasi/publications.html.

12. Please say in figure captions if pole figures are equal area or equal angle. I think they are equal area from the shapes of maxima (the projection affects shape analysis of maxima).

Response: We changed the captions in figure 2 and 4 accordingly.

13. It would be really cool to see a radial section of the sample: to see how microstructure changes with strain in a single sample (e.g. see (King et al., 2011). I'm not suggesting this is needed for this paper- just something cool to do.

Response: This response is similar to the one given to reviewer 1. One axial section was measured with EBSD, but due to the coarse grain size discussed above, it was impossible to fractionate the section in different segments with 'supposed' constant finite shear strains and still have a high enough number of measurements to obtain representative estimates of the CPO intensity. Therefore, we could not use radial sections to estimate the CPO evolution with increasing strain.

14. There are a few key references on experimental shear of ice that are missing and should be cited. These include (Budd et al., 2013; Li et al., 2000; Wilson and Peternell, 2012).

Response: We included these in the revised version of the manuscript.

Added text (pp.3 L.2): "A similar evolution was observed in more recent shear experiments on artificial ice polycrystals by Li et al. (2000) and Budd et al. (2013), as well as by Wilson and Peternell (2012) which analyzed the influence of the initial CPO and the importance of recrystallization processes on the CPO evolution. "

15. There are several published papers that show a lack of CPO change in rocks during annealing. Some of these should be cited.(Augenstein and Burg, 2011; Heilbronner and Tullis, 2002; Ree and Park, 1997). I know there are others in calcite and olivine but can't find them just now.

Response: We thank the reviewer and included these in the revised version of the manuscript.

16. Throughout this paper the term "texture" is used with the meaning common in metallurgy and materials science. There is a very small community of geoscientists who use "texture" in this way and no glaciologists that I know of. For the vast majority of the geoscience community "texture" means the spatial relationships of phases and grains

and their internal structures. To most geoscientists, texture is what you would see down a microscope (in a petrographic examination for example) and is broadly synonymous with the term microstructure. The terms "crystallographic preferred orientation" (CPO: which you use in the intro) or "lattice preferred orientation" (LPO) are much better as they are explicit. If you want this paper to have wider readership/ uptake, remove the word texture throughout and replace with CPO. It is also worth (in the intro) relating this terminology to the word "fabric" and/or the acronym "COF" (crystal orientation fabric) as commonly used in glaciology. I avoid using the term fabric (except in explanations of how terminology matches up) as metallurgists use this term to mean microstructure.

Response: We agree with the reviewer and replaced texture by CPO in the entire manuscript.

17. It is not really clear what are the observations you use to constrain the dimensions of the bulging nucleus.

Response: We use the similarity in the microstructures observed in the present experiments with those described in Chauve et al. (2017) to suggest that bulging associated with formation of low angle grain boundaries may be an efficient nucleation mechanism. The increase in the c-axis component in the WBV of the low angle boundaries in the first 100 $\mu$m from the grain boundary is an indicator of the presence of sub-grain boundary loops with c-component GNDs, which as described in Chauve et al. (2017) play an essential role in closing the bulges. Thus the width we extrapolate for a maximum bulging nucleus of 100$\mu$m, which is controlled by the length scale over which the stresses are high enough to activate the hard non-basal slip systems and close bulging grains.

18. I don't follow the discussion related to nucleation in the section where the annealing is discussed. Grain size increases during the annealing so nucleation is unnecessary. If you are talking about relationships that might be relevant to nucleation prior to the annealing then this needs to be made clear.

Response: We do not suggest that nucleation necessary occurs during annealing. We just highlight the fact that grain boundary migration during annealing does not drastically modify the texture. Therefore, new grains present prior to annealing, with low defect density and greater chance to grow through GBM, must have had orientations close to M1 and M2. This suggest bulging as the dominant mechanism for nucleation at the conditions of our experiments, as it tends to create grains with a closer orientation to the parent grain. We rephrase this section to make our point clearer.

Added text : (pp.17 L.15) "This suggest bulging as the dominant mechanism for nucleation at the conditions of our experiments (prior to annealing), as it tends to create grains with a closer orientation to the parent grains. Furthermore we report an offset angle between the M1 and M2 submaxima of 17.4âŮę greater than what would be expected for a perfect symmetry to the finite extension direction. Other experiments, which didn't undergo annealing show a difference in angle of only 1-3âŮę with the perfect symetry. In the TGI0.71 annealed sample, M2 is much closer to M1, as would be expected for a higher finite shear strain. This could be interpreted as a sign of preferential growth of bulging nucleii with orientation closer to M1 than the bulk CPO of the sample before annealing "

19. Bulges cut off by rotation of a subgrain boundary was first suggested (described from see through experiments) by Janos Urai (I think). You should reference (Urai et al., 1986).

Response: We included this reference in the discussion (pp.17 L. 33) of the revised version of the manuscript.

20. Spontaneous (random) nucleation? I have a problem with this - it is a bit of magic with no physically realistic explanation.

Response: Spontaneous random nucleation was first hypothesized by Duval et al. (2012), who proposed that the energy for nucleation is provided by internal stress field associated with dislocations pile-ups. CPO data corroborating the existence of this process as a secondary nucleation mechanism, the dominant one being the association of bulging and subgrain rotation, were presented by Chauve et al. (2017).

21. The conditions of your experiments are not close to those in cold glaciers and ice streams (page 18, line 5). Your slowest transient strain rate is 2.7E-7s-1 which corresponds to a 100m thick shear zone having a velocity difference across it of 850m/yr. The tertiary strain rate in your high strain experiment corresponds to ∼2700m/yr difference across a 100m shear zone. I'm not so familiar with temperate glaciers but such shear rates do not exist in polar ice sheet systems eg (Bons et al., 2018; Rignot et al., 2011). Even fast ice stream shear margins max out below 1E-9s-1 (Bindschadler et al., 1996; Jackson, 1999; Jackson and Kamb, 1997). The strain rate has a significant effect on the microstructure and the CPO (Hirth and Tullis, 1992; Qi et al., 2017; Tullis, 1972): increasing strain rate has a comparable effect to decreasing temperature. It is not possible to do an experiment to significant strain at natural conditions. Instead experiments need to provide scaling relationships that allow us to predict the effects of T, strain rate (stress) etc on rheology and CPO/microstructure (with the complication that there are feebacks where CPO/microstructure affect the rheology).

Response: We thank the reviewer for this useful comment and changed the sentence in the revised version of the manuscript as:

Changed text (pp.19. L.28) "The experiments, performed at high temperature, up to shear strains of 2, favored dynamic recrystallization observed in natural conditions with slower strain rates such as cold glaciers, ice streams, and some deep ice core areas."

Please also note the supplement to this comment:
https://www.the-cryosphere-discuss.net/tc-2018-213/tc-2018-213-AC2-supplement.zip

---

## Author Response (AR2)

Response to referee #2 David Prior:

*This remains an excellent contribution. I'm pleased to see that a lot of comments from all reviewers (official and not) have been taken on board and the paper is much improved because of this. There are a lot of minor errors in the English in the revised version and a few places where the text needs greater clarity. The conclusions are disappointing as they fail to summarise some of the key observations related the CPO with increasing strain (M1,M2, <a> and <m>).*

We thank the referee for the in-depth review initially provided that helped a lot the quality of the manuscript.
We added some details and a bullet point in the conclusions to underline better the key observations related to the evolution of the CPO with strain.

*Some comments on responses to reviewers:*

*Scientific Discussion 2: The new supplementary material is all very useful. Could you add bigger and better resolution photo and drawing of the apparatus. The words on this slide can take up less space.*

We reorganized the supplementary material to address this comment.

*11. What you say here is not correct. It's possible to have a sublimation condition in a cold room where air and sample are at the same temperature and the partial pressure of water in the air can be below the pressure that is in equilibrium with the ice. This is pretty much impossible if the ice is cold (even if only -20C) and the room at ~20C, even if the room is de-humidified or (as in our case) has a constant flow of nitrogen. You might not see the frost and it only needs a few tens of nm to be a problem. I'm sure that your frost goes when you pump down the chamber. So rather than arguing about this just include what temperature the sample is at when it is put into the SEM. This means others can reproduce (sort of) what you have done. If the sample is warmer than ~-80 it is highly likely to sublime during the pump down to high vacuum (pressure cycling).*

We changed the phrasing in the corresponding section of the manuscript to just specify the loading temperature at -60ºC.

*Comments/ corrections for the manuscript: in page/ line number order.*

*Page 1. Line 1-2. "... reproduce simple shear conditions close to those encountered in ice streams and ..". This phrasing is misleading as it can imply all conditions (T, strain rate etc). I would rephrase with something that relates specifically to the kinematics "to reproduce the simple shear kinematics that are believed to dominate in ice streams and .."*

Done.

*Page 1. Line 6. "in natural setups" is not great English. How about "in naturally deformed ice"*
Done.

*Page 1. Line 7. Electron BackScattering Diffraction is not a common full version for EBSD. Most literature and books use Electron Backscatter Diffraction.*
Done.

*Page 1. Line 9. Add the word "an" to make "...form under torsion of an initially ..."*
Done.

*Page 2. Line 11. references should be plural.*
Done.

*Page 2. Line 34. As you are referring to the pioneering studies that show this for the first time you should include Kamb 1972 in this list. This is the first substantial and easily available piece of work.*
Done.

*Page 3 Line 21-24. Not quite as simple as this. The double maximum is there in the Llorens et al models (see discussion in Qi et al 2019). None of the models really produce the interlocking grains (including the Etchecopar model: it can't really). The later discussion on models now has a much better and more scientific tone than the original. This section effectively picks out one model to point out some problems that are common to many models. That's not very helpful. Reducing this paragraph to say simply that we need good constraints to test models, without pointing out specific problems is better. Not that interlocking grains is a high temperature/ low strain rate thing. You don't get that at lower T or fast rates (e.g. the -20,-30 experiments in Qi et al).*
We rephrase this part of the manuscript to address this comment.

*Page 4. Line 23. Replace "Supplementary" with "Supplementary material".*
Done.

*Page 4. Line 20. Add Li et al 2000 to the Swift 1947 citation. Li et al relates directly to ice.*
Done.

*Page 5 Line 18. PETERNELL in caps?*
Fixed.

*Page 5 Line 19. "were operated" is not good English as the object of the sentence is the measurments not the machine. "All optical CPO measurements were conducted at -7..." is better.*
Done.

*Page 5 Line 27. Replace "Supplementary" with "Supplementary material".*
Done.

*Page 5 around Line 30. Please add what temperature (or range of temperatures) the ice is when it is put in the SEM. See my comment to your response about sublimation later.*
Done.

*Page 6 Line 3. Just say at approximately -60C. It is more complicated than your statement makes out as the pressure of interest is the partial pressure of H20 in the SEM chamber rather than the vacuum pressure, which is the addition of the partial pressures of the gasses involved (itrogen and water).*
Done.

*Page 7 Line 23. Replace "..less than 25 per radii." With "..fewer than 25 grains per radii."*
Done.

*Page 10 Line 22. Replace "The CPO seems not modified.." with "The CPO does not seem to be modified.."*
Done.

*Page 10 Line 30. "Its almost disappearance" is poor English. How about : "The nearly complete disappearance of M2 correlates…"*
Done.

*Caption to fig 3 (and elsewhere). I presume the J index here relates just the c-axes; what Dave Mainprice's old software called pfJ (pole figure J). As such it is not identical to most peoples understanding of the J index (for full crystallograophic orientations). It would be good to make this clear.*
We added details at several place in the manuscript to make clear that we are referring to the c-axis J index.

*Page 11 Line 5. "allows to perform" is poor English. This sentence could be: "The access to the full crystallographic orientations enables more complete grain segmentation based on misorientation data."*
Done.

*Page 11 Line 10. J index- see above (caption to fig 3).*
Done.

*Fig 4b. <a> and <m> axes. Why not use the colour scale for <a> and <m> as applied for The gamma 1.96 sample for all of the samples. Then one can see how these evolve? The current formatting of this figure is very confusing for the reader.*
Done.

*Fig 4b. Rotated view of gamma 1.96 sample. The rotation is not correct. In the left hand reference frame x is between the two <c> sub-maxima of the elongated overall maximum. In the right hand reference frame x lies on one of the sub-maxima. The y axis is differently aligned*

*relative to the <a> and <m> sub maxima in the right and left hand reference frames. It looks like the data on the right hand side needs rotating about 20 degrees around x followed by ~ 20 degrees around y?*

We thank the reviewer to have spotted this, this was a mistake where some rotation on the X axis was applied. The new version has only 90º rotation on the Y axis.

*Page 13 Line 7. "allows to" is poor English. How about "WBV analysis allows calculation of GNDs and definition of the relative…"*

Done.

*Page 14 Line 14. "almost no-sub-grain boundaries". This wording does not reflect the truth- it's rather misleading. This is what I see. "Both the undeformed sample and the annealed sample have much lower low angle boundary densities than the other samples. The undeformed sample map shows low angle boundaries very close to high angle boundaries and as unconnected pixels and small segments within grains. These are best explained as low angle misindexing errors. Most grains in the annealed sample contain no low angle boundaries. There are ~ 4 grains that contain low angle boundaries that are similar to the more extensive low angle boundaries observed in the defomed but not annealed samples."*

We rephrase this part of the manuscript including the wording of the referee. We thank him for the helpful comment.

*Page 14 Line 15. The sentence that starts on this line is really difficult to understand. I think if you remove "with a significant ||WBV||" from line 15 it is much clearer.*

We rephrase this part of the manuscript to address this comment.

*Page 14 Line 18. Replace "lower statistics" with "smaller sample size".*

Done.

*Page 14 Line 24. Replace "poor statistics" with " the small sample size".*

Done.

*Page 15 Line 2-4. This says nothing. Just remove these three lines.*

Done.

*Page 15 Line 8. Replace "apparition" (means a ghost or a miracle!) with "appearance".*

Done.

*Page 17 Line 7. Replace "apparition" with "appearance".*

Done.

*Page 17 Line 15. "suggest" should be "suggests"*

Done.

*Page 17 Line 17. I'm not sure there is any evidence that bulging gives closer orientations to parents. If the "break off" mechanisms is subgrain rotation then the two mechanisms will give the same relationship.*

As it is stated in the manuscript, this is just an hypothesis we are making in our interpretation of the results. We are conscious that this not a solved issue, and we feel that this is clear for the reader in the current version of the manuscript.

*Page 17 Line 35. Replace "amount" with "number".*

Done.

*Page 18 Line 6. The Urai et al reference is wrong here and in the list. Probably the reference in citation software is wrong. Full reference should be:*
*Urai, J. L., Means, W. D., and Lister, G. S., 1986, Dynamic recrystallization of Minerals, in Hobbs, B. E., and Heard, H. C., eds., Mineral and Rock Deformation (Laboratory Studies), Volume 36, p. 161-200.*
*Another key ref here that I had forgotton about is:*
*Stipp, M., and Kunze, K., 2008, Dynamic recrystallization near the brittle-plastic transition in naturally and experimentally deformed quartz aggregates: Tectonophysics, v. 448, no. 1-4, p. 77-97.*

Done.

*Page 18 Line 15 to Page 19 line 10. You have reverted to using the term texture rather than CPO here. Probably elsewhere in the manuscript as well. Do a search to ensure terminology is the same in the bulk of the paper.*

We corrected this in the new version of the manuscript.

*Page 18 Line 17. Etchecopar does not use the term polygonazation and does not allude to this process. He talks about grains breaking. You should make it clear that thinking of this as polygonization is a reasonable extension of the Etchecopar idea. You could extend the phrase in brackets to "(Etchecopar defines this as grain breaking but grain polygonization would have the same kinematic effect)".*

Done.

*Page 19 Line 18. Replace "M2 CPO" with "M2 maximum".*

Done.